# LUMINA-OMNILV: A UNIFIED MULTIMODAL FRAMEWORK FOR GENERAL LOW-LEVEL VISION

## ABSTRACT

We present **Lunima-OmniLV** (abbreviated as **OmniLV**), a universal multimodal multi-task framework for low-level vision that addresses over 100 sub-tasks across four major categories, including image restoration, image enhancement, weak-semantic dense prediction, and stylization. OmniLV leverages both textual and visual prompts to offer flexible, user-friendly interactions. Built on Diffusion Transformer (DiT)-based generative priors, our framework supports arbitrary resolutions — achieving optimal performance at 1K resolution — while preserving fine-grained details and high fidelity. Through extensive experiments, we demonstrate that separately encoding text and visual instructions, combined with co-training using shallow feature control, is essential to mitigate task ambiguity and enhance multi-task generalization. Our findings also reveal that integrating high-level generative tasks into low-level vision models can compromise detail-sensitive restoration. These insights pave the way for more robust and generalizable low-level vision systems.

## 1 INTRODUCTION

The rapid evolution of large-scale foundation models has revolutionized artificial intelligence, demonstrating remarkable generalization and multi-task capabilities across various domains. Unified frameworks such as GPT-4V (Achiam et al., 2023), InternVL (Chen et al., 2024h;f;g), Flamingo (Alayrac et al., 2022), OmniGen (Xiao et al., 2024), and OneDiffusion (Le et al., 2024b) have showcased impressive performance by leveraging large-scale pretraining on multimodal datasets. These models excel in semantic-driven high-level vision tasks, such as image classification, image understanding, visual generation and editing. In contrast, the development of unified models for low-level vision remains largely fragmented and underexplored.

Low-level vision encompasses a broad spectrum of tasks, including image restoration (Dong et al., 2015; Zhang et al., 2017; Chen et al., 2024d; 2023; Lin et al., 2024b; Yu et al., 2024a; Lin et al., 2025) , image enhancement (Cai et al., 2023; Zamir et al., 2020; Chen et al., 2021c;b; Wang et al., 2019), style transfer (Gatys et al., 2016; Huang & Belongie, 2017), and weak-semantic dense prediction (Yang et al., 2024a; Kirillov et al., 2023; Ravi et al., 2024) (e.g., edge detection, depth estimation, normal map estimation). Unlike high-level vision tasks that rely on predefined semantic understanding, most low-level vision tasks do not require explicit object-level reasoning. Instead, they focus on pixel-level fidelity, fine-grained texture reconstruction, and feature extraction. This distinction makes the unification of low-level vision tasks particularly challenging, as different tasks often operate in vastly different output domains.

Existing approaches to low-level vision remain limited in generalization, usability, and scalability. Task-specific models (Chen et al., 2023; Li et al., 2022a) are designed to handle a single task (e.g., denoising, deblurring, super-resolution), requiring extensive model redesigning and retraining to adapt to new tasks. All-in-one restoration models, such as AirNet (Li et al., 2022a), PromptIR (Potlapalli et al., 2023), and OneRestore (Guo et al., 2024), integrate multiple restoration tasks within a single framework, yet remain restricted to in-domain restoration, unable to generalize to cross-domain tasks such as feature extraction or style transfer. Visual-prompt-based models, such as PromptGIP (Liu et al., 2023a) and GenLV (Chen et al., 2024e; 2025), extend to cross-domain tasks using image prompt pairs, but require carefully crafted prompts, making them less intuitive and user-friendly compared to text-driven interaction. Furthermore, many existing methods operate only on fixed-resolution images, severely limiting their flexibility and real-world applicability.

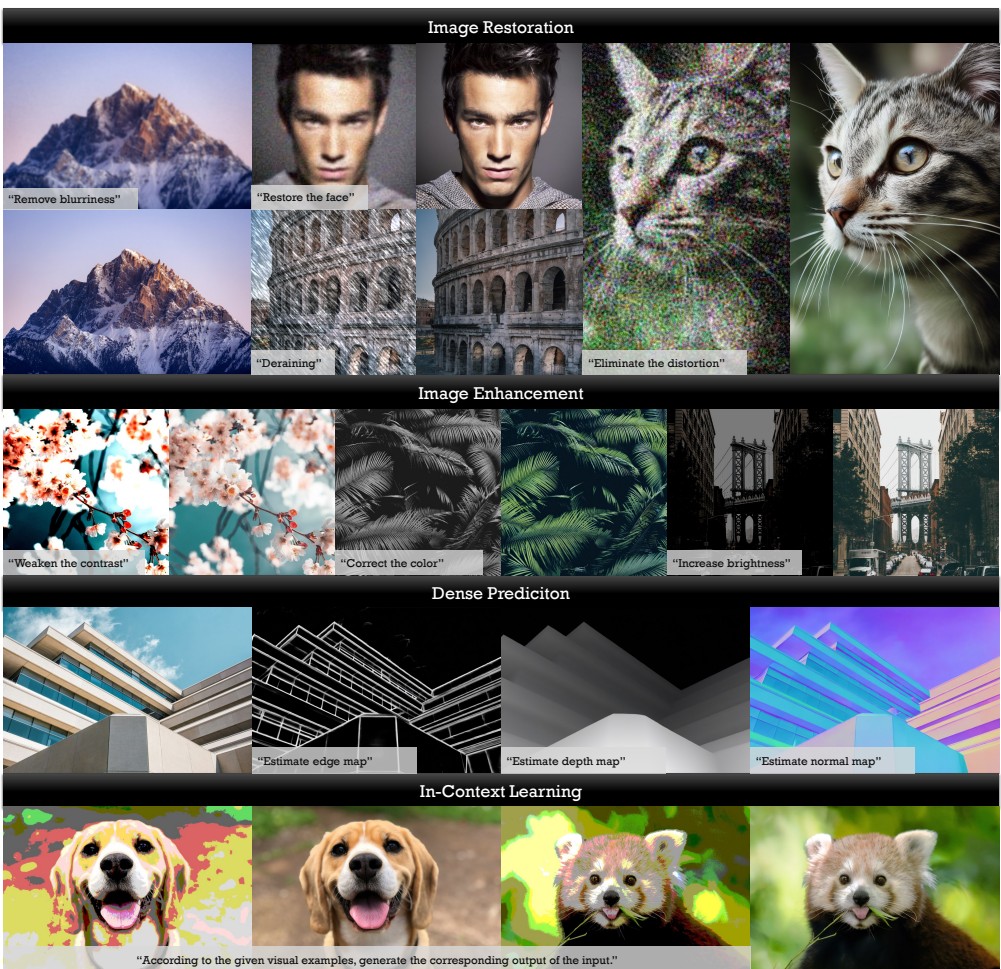

Figure 1: **Illustration of OmniLV's versatile capabilities.** As a universal framework, OmniLV is capable of handling a wide variety of low-level vision tasks within a single model, which adapts to diverse input-output domains and generates high-fidelity results.

To summarize, high-resolution image processing still remains challenging, leaving ample room for improvement in task adaptability.

Given the inherent complexity and diversity of low-level vision, developing a truly universal model must handle multiple task domains while reliably preserving fine-grained details and high fidelity. A key requirement for such a model is flexible interaction mechanisms. While text-based instructions offer a convenient and intuitive way to specify tasks (e.g., "remove noise from this image", "enhance brightness", and "estimate the Canny edge"), certain tasks — such as style transfer — are difficult to define using text alone. Visual prompts, provided in the form of exemplar image pairs, provide an effective alternative by allowing the model to infer complex, task-specific transformations through visual analogy. Thus, an ideal general low-level vision model should integrate both textual and visual prompts for versatile and user-friendly task execution.

To address these challenges, we propose **OmniLV**, a universal multimodal multi-task framework for low-level vision, capable of handling over 100 sub-tasks via both textual and visual prompts. Built on Diffusion Transformer (DiT)-based generative priors (Ho et al., 2020; Rombach et al., 2022; Peebles & Xie, 2023; Esser et al., 2024), our model significantly improves generalization and output quality across tasks. Unlike prior models constrained to fixed resolutions, our framework supports arbitrary resolutions, achieving optimal performance at 1K resolution. We systematically explore multimodal fusion strategies and propose a simple yet effective design that prevents task misinterpretation issues. Fig. 1 presents the versatile capabilities of OmniLV.

Throughout the development of OmniLV, we have gained several key insights that shape the design of a robust and generalizable low-level vision model. First, we find that separately encoding text-based and visual instructions is crucial for preventing task ambiguity, as naive fusion can lead to task misinterpretations (Sec. 3.1.2). Additionally, co-training the base model with shallow feature control proves to be an effective strategy for enhancing multi-task generalization (Sec. 3.1.3). Furthermore, incorporating high-level generative or editing tasks into a low-level vision model significantly compromises fidelity, particularly in detail-sensitive restoration tasks (Sec. 4.2). These findings highlight the need for dedicated multimodal architectures tailored for low-level vision tasks.

In summary, our work makes the following key contributions. (1) We present the first unified multimodal framework capable of handling four major low-level vision categories (over 100 sub-tasks) through both text and image interactions. (2) We introduce an effective multimodal fusion mechanism that aligns text and image prompts, mitigating task misalignment issues. (3) We provide new empirical insights into challenges of building multi-task low-level vision generalists, revealing how the integration of high-level generative and editing tasks can adversely impact fidelity-critical tasks.

## 2 RELATED WORK

### 2.1 IMAGE RESTORATION WITH GENERATIVE PRIOR

Diffusion-based methods have emerged as a robust framework for image restoration, converting degraded inputs into high-quality outputs through reverse denoising. Several key works illustrate the versatility of this approach (Wang et al., 2024a; Lin et al., 2024b; Yang et al., 2024b; Yu et al., 2024a; Ai et al., 2024; Wu et al., 2024; Chen et al., 2024a; Yue et al., 2024). StableSR (Wang et al., 2024a) leverages the generative priors of pre-trained text-to-image diffusion models for blind super-resolution, employing a time-aware encoder and feature wrapping to balance quality and fidelity while accommodating arbitrary resolutions. DiffBIR (Lin et al., 2024b) uses a two-stage pipeline where the first stage reduces degradations and the second stage employs a latent diffusion model to generate missing details, proving effective in denoising and face restoration. PASD (Yang et al., 2024b) extends the Stable Diffusion framework for realistic super-resolution and personalized stylization by integrating a pixel-aware mechanism that improves both resolution precision and style adaptability. SUPIR (Yu et al., 2024a) scales up large diffusion models such as StableDiffusion-XL, incorporating a trained adapter and a massive high-resolution dataset to enable text-guided, photorealistic restoration in complex scenes. However, the limitation of these approaches is that they are confined to image restoration tasks and cannot address other challenges in low-level vision.

### 2.2 ALL-IN-ONE GENERATIVE MODELS

Developing all-in-one models is an exciting yet challenging pursuit. In the realm of image generation, various studies have sought to build versatile systems (Wang et al., 2024b; Le et al., 2024a; Xiao et al., 2024; Lin et al., 2024a; Han et al., 2024; Mao et al., 2025). For example, OmniGen (Xiao et al., 2024) encodes text and images into a unified tensor, utilizing causal attention for text tokens and bidirectional attention for image tokens. Pixwizard (Lin et al., 2024a) introduces task-specific embeddings for image editing and understanding, while ACE (Han et al., 2024; Mao et al., 2025) offers a conditioning module that accepts diverse input images and processes them concurrently with a transformer. UniReal (Chen et al., 2024c) employs a video generation framework that treats images as individual frames, providing a universal solution for various image generation and editing tasks.

Despite these advances, most of these approaches focus on image generation and editing, leaving universal models for low-level vision relatively unexplored. Visual prompt-based approaches (Liu et al., 2023a; Chen et al., 2024e; 2025) tackle cross-domain tasks by utilizing pairs of image prompts. However, their dependence on meticulously crafted prompts renders them less intuitive and user-friendly compared to text-driven alternatives. Moreover, many current methods are restricted to fixed-resolution outputs, limiting their practical applicability.

## 3 METHOD

### 3.1 BUILDING OMNILV STEP-BY-STEP

In this section, we detail the key design choices and learned insights in developing a universal low-level vision model, outlining our step-by-step thinking process.

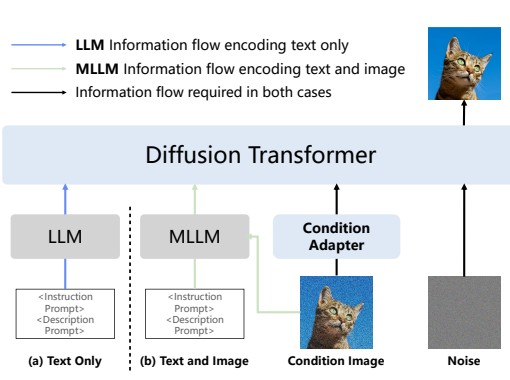

Figure 2: Comparison between MLLM guided and LLM guided framework.

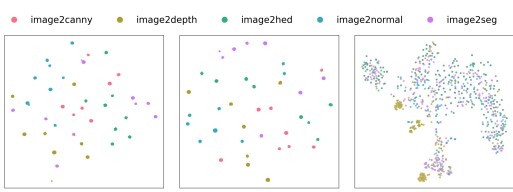

| | Compression | | Quantization | | Noise | | Inpainting | |
|---|---|---|---|---|---|---|---|---|
| | PSNR↑ | MUSIQ↑ | PSNR↑ | MUSIQ↑ | PSNR↑ | MUSIQ↑ | PSNR↑ | MUSIQ↑ |
| Addition | 21.92 | **56.31** | **18.72** | **55.71** | 22.34 | **59.31** | **19.94** | **56.96** |
| Concat | **21.93** | 55.99 | 18.18 | 55.11 | **22.35** | 57.13 | 19.86 | 55.95 |

Table 1: Ablation study on whether to use addition or concatenation in in-context learning.

Figure 3: t-SNE visualization of the feature space of LLM and MLLM. Each dot represents a task instruction.

### 3.1.1 PRELIMINARY: BASE MODEL SELECTION

Unlike most foundational image restoration models (Ho et al., 2020; Rombach et al., 2022; Peebles & Xie, 2023; Esser et al., 2024) that are trained from scratch using deterministic regression objectives, we leverage a pre-trained text-to-image diffusion model as a strong initialization. Pre-trained diffusion models (Esser et al., 2024; Zhuo et al., 2024; Gao et al., 2024; Xie et al., 2024; Labs, 2024), trained on billions of images, offer rich visual priors that enhance generalization, support diverse resolutions and aspect ratios, and effectively capture the uncertainty inherent in multi-task image restoration. These properties contribute to a more robust and versatile low-level vision model.

For our base model, we initialize with Lumina-Next (Zhuo et al., 2024; Gao et al., 2024), a flow-based diffusion transformer that introduces several architectural improvements over traditional DiT-based models (Peebles & Xie, 2023), including 2D Rotary Positional Encoding, QK Normalization, and Sandwich Normalization. Additionally, Lumina-Next adopts a flow-matching formulation, improving training stability and accelerating convergence. To adapt this model for general low-level vision, we introduce a condition adapter that integrates inputs to enable effective task conditioning, which is illustrated in subsequent sections. The modified model is trained using a flow-matching loss to learn a conditional time-dependent velocity field, facilitating the transformation between noisy and clean image distributions. Please refer to Appendix C.3 for details of training loss.

### 3.1.2 ENCODING MULTIMODAL INFORMATION

Given an input image $x$, our goal is to generate the target image using both textual instructions and in-context visual exemplars. We explore two different encoding strategies: (1) **Separate encoding**, where text prompts are processed using a large language model (LLM), while visual exemplars are encoded independently. (2) **Unified encoding**, where both text and visual inputs are fused within a multimodal language model (MLLM). Fig. 2 illustrates the architectural differences between these two approaches. While unified encoding benefits from parameter efficiency and leverages cross-modal correlations, we observe that it introduces critical limitations especially when applied to dense prediction tasks. Specifically, multimodal encoders often misinterpret task instructions, leading to inconsistencies in generated outputs. To better understand this issue, we visualize the encoded feature distributions in Fig. 3. Our findings indicate that mixing text and image prompts within a single encoder leads to severe task ambiguity. Since visual tokens dominate the shared feature space, text-based instructions often get overshadowed, leading to misalignment and incorrect outputs. Please refer to Fig. 10 in the Appendix for an illustration of task misalignment.

Based on these observations, we adopt a separate encoding strategy: text instructions are processed via an LLM, while image exemplars are encoded using a visual VAE. This ensures clearer task separation, preventing interference between textual and visual guidance, and improves task accuracy across a vast number of low-level vision tasks.

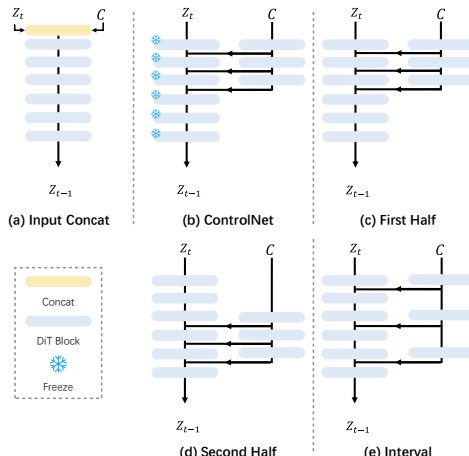

**Figure 4: Overall framework of OmniLV.** First, input images are encoded into latent space by VAE encoder. Then, we patchify the image latent and noise latent into visual tokens. Optionally, in-context pairs can be added to visual tokens to handle complex scenarios. At the same time, the instruction prompt and description prompt are processed by Gemma2B. Finally, we decode the denoised results to get the desired output images.

Figure 5: Schematic structures of five different variants.

Table 2: Ablation study on condition integration.

| | Position | Train DM? | SIDD | | RealBlurJ | | SR | |
|---|---|---|---|---|---|---|---|---|
| | | | PSNR↑ | MUSIQ↑ | PSNR↑ | MUSIQ↑ | PSNR↑ | MUSIQ↑ |
| (a) | Input | ✓ | 32.40 | 21.91 | 22.98 | 51.66 | 22.89 | **56.89** |
| (b) | First Half | ✗ | 25.52 | 23.53 | 22.28 | 44.41 | 21.11 | 50.44 |
| (c) | First Half | ✓ | **34.09** | **23.96** | **24.05** | **57.42** | **22.93** | 56.72 |
| (d) | Second Half | ✓ | 29.60 | 23.38 | 23.15 | 54.35 | 22.96 | 56.60 |
| (e) | Interval | ✓ | 34.07 | 23.06 | 22.77 | 53.50 | 22.80 | 56.38 |

Table 3: Effects of Various Prompt Formats. Our approach supports text prompts, visual prompts, or a combination of both.

| Text | Visual | Blur | | Noise | | Contrast Adju. | | Saturate Adju. | |
|---|---|---|---|---|---|---|---|---|---|
| | | PSNR↑ | MUSIQ↑ | PSNR↑ | MUSIQ↑ | PSNR↑ | MUSIQ↑ | PSNR↑ | MUSIQ↑ |
| ✓ | ✗ | **22.57** | 68.95 | **23.53** | 69.23 | **20.90** | 69.95 | **21.79** | 70.90 |
| ✗ | ✓ | 21.99 | 67.59 | 22.71 | 67.66 | 20.09 | 66.58 | 20.14 | 68.08 |
| ✓ | ✓ | 22.50 | **68.99** | 23.07 | **69.37** | 20.50 | **70.09** | 21.00 | **70.91** |

Table 4: Ablation study for the training data.

| High-semantic? | Blur | | Noise | | Contrast Adju. | | Saturate Adju. | |
|---|---|---|---|---|---|---|---|---|
| | PSNR↑ | MUSIQ↑ | PSNR↑ | MUSIQ↑ | PSNR↑ | MUSIQ↑ | PSNR↑ | MUSIQ↑ |
| ✗ | **21.13** | **58.78** | **22.34** | 57.39 | **19.03** | **56.95** | **18.64** | **56.60** |
| ✓ | 20.97 | 57.06 | 22.18 | **59.31** | 18.76 | 55.80 | 18.58 | 56.55 |

### 3.1.3 DESIGN CHOICES OF CONDITION INTEGRATION

Integrating condition images into diffusion models is commonly achieved through two primary approaches: (1) **Feature Injection:** A trainable adapter injects feature maps into a frozen diffusion model (Zhang et al., 2023b; Mou et al., 2024). (2) **Input Concatenation:** Condition images are concatenated with inputs, and the entire model is fine-tuned. These designs have been widely used in in-domain single task (e.g. image restoration, canny2image), achieving remarkable results (Yu et al., 2024a; Lin et al., 2024b). To systematically investigate condition integration strategies for general low-level vision tasks, we conduct comparative experiments evaluating different design choices (see Fig. 5). Our findings, summarized in Tab. 2, are as follows: (1) Training only the adapter is suboptimal (settings (b) & (c)), indicating that fine-tuning the base model is necessary for adapting generative priors to diverse low-level tasks. (2) While input concatenation is efficient(setting(a)), adding additional parameters to process the condition image enhances performance (setting (c)), suggesting that explicitly modeling condition images helps extract more relevant structural and con-

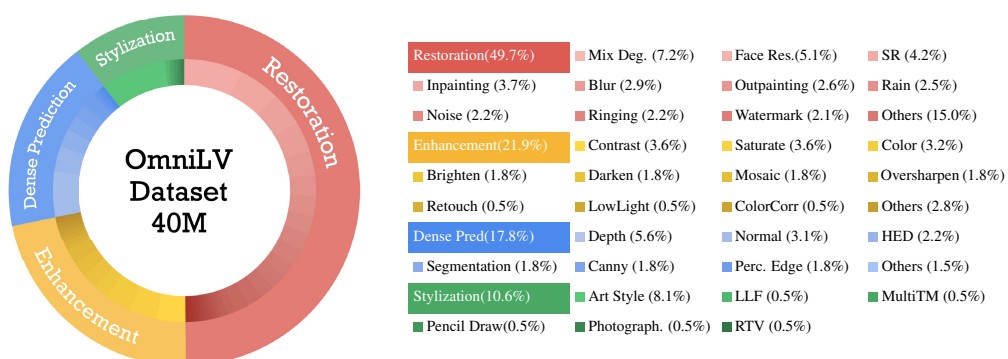

Figure 6: OmniLV dataset distribution with main categories.

textual information. (3) The injection position significantly influences the performance (settings (c), (d), & (e)). Integrating condition information in the first half of the network leads to better results, likely because early-stage modulation ensures stronger feature guidance throughout the process.

Based on these findings, we propose a co-training condition adapter, which jointly optimizes the adapter and base model. Unlike ControlNet-like architectures, which keep the base model frozen, our approach ensures deeper feature alignment, improving multi-task generalization and fidelity.

### 3.1.4 ENABLING IN-CONTEXT LEARNING

While text prompts can effectively guide tasks, many low-level vision tasks (e.g., stylization) require precise visual instructions that are difficult to express linguistically. To address this, we compare two paradigms for visual prompt integration: (1) **Input Concatenation** (Chen et al., 2024c; Xiao et al., 2024; Wang et al., 2024c), where visual prompts are concatenated along the token dimension:

$$\mathbf{H}_{\text{fused}} = [\mathbf{H}_{\text{img}}; \ \mathbf{H}_{\text{prompt}_1}; \ \ldots; \ \mathbf{H}_{\text{prompt}_n}], \tag{1}$$

where $\mathbf{H}_{\text{img}}$ and $\mathbf{H}_{\text{prompt}_i}$ denote latent representation of input image and latent representation of i-th visual prompt, and $\mathbf{H}_{\text{fused}}$ denotes the combined latent representation. (2) **Projection-Addition** Wang et al. (2023c), which employs lightweight projectors to align visual prompts with the latent space before summation:

$$\mathbf{H}_{\text{fused}} = \mathbf{H}_{\text{image}} + \sum_{i=1}^{n} \phi_i(\mathbf{H}_{\text{prompt}_i}), \tag{2}$$

where $\phi(\cdot)$ denotes linear projectors. Fig. 4 illustrates the architectural differences between these two approaches, where the concatenation method can be seen as a variation where the "Projector-Addition" module is replaced with a Concatenation operation. Tab. 1 presents the quantitative comparison, demonstrating that projection-addition outperforms input concatenation across different tasks. This suggests that projection-based alignment better preserves task-relevant information.

**Final Architecture.** Based on these insights, we design the final architecture of OmniLV, as illustrated in Fig. 4. Our approach unifies diverse low-level vision tasks while ensuring strong multi-modal conditioning and in-context learning capabilities.

### 3.2 LARGE-SCALE OMNILV DATASET

To build a universal low-level vision model, we construct a large-scale multi-task dataset containing 40 million instances over 100 sub-tasks across four major domains: image restoration, image enhancement, dense prediction, and stylization. The main categories and distribution of OmniLV dataset are illustrated in Fig. 6. The dataset is sourced from publicly available collections and synthetically generated pairs, with additional high-quality data created through internal pipelines.

**Image Restoration.** The restoration dataset covers 23 major tasks with a total of 45 sub-tasks, addressing various degradation types such as motion blur, noise, and weather-induced distortions. It consists of both real-world degraded images and synthetic degradation pairs, carefully processed alongside high-quality ground truth images to ensure realism and diversity.

Table 5: Quantitative comparison on restoration tasks. Red and blue colors represent the best and second best performance, respectively, excluding specialized models. All values are reported as PSNR↑ / MUSIQ↑. For specialized models, if a model achieves the best value, the corresponding number is highlighted in **bold**.

| Category | Method | Deblur | | Compression | Denoise | | Derain | | Desnow | BIR | Face |
|---|---|---|---|---|---|---|---|---|---|---|---|
| | | OLV-T(6 types) | RealBlur-J | OLV-T(2 types) | OLV-T(6 types) | SIDD | Synthetic | Rain1400 | Snow100K-L | DIV2K | CelebA |
| Specialized | X-Restormer | 21.18/45.17 | 26.57/50.41 | – | **27.19**/63.67 | 31.95/22.04 | 27.10/71.34 | 32.35/70.34 | – | – | – |
| | MPRNet | 20.33/43.35 | 26.51/48.45 | – | 24.53/45.66 | 39.63/22.34 | 25.16/69.40 | 32.04/69.98 | – | – | – |
| | MAXIM | 21.39/44.13 | **29.99**/55.68 | – | 24.75/49.30 | **39.68**/22.32 | 25.82/71.15 | 32.25/70.27 | – | – | – |
| | DiffBIR | – | – | – | – | – | – | – | – | **22.77**/67.01 | – |
| | GFPGAN | – | – | – | – | – | – | – | – | – | **25.80**/69.76 |
| | CodeFormer | – | – | – | – | – | – | – | – | – | 25.15/**75.55** |
| All-in-One Restoration | X-Restormer | 21.44/39.73 | 26.23/38.84 | – | 25.96/62.42 | 24.06/20.84 | 23.28/69.00 | 32.12/70.26 | – | – | – |
| | DA-CLIP | 19.94/34.98 | 18.82/39.22 | – | 22.99/44.89 | 26.40/29.25 | 23.15/53.18 | 26.44/67.78 | – | – | – |
| | AutoDIR | 20.09/45.07 | 19.10/49.63 | – | 26.46/57.80 | 22.19/28.72 | 25.33/64.59 | 26.21/70.75 | – | – | – |
| Visual-Prompt-based | Painter | 17.05/28.74 | 15.37/28.79 | 17.84/34.43 | 18.04/37.11 | 38.65/21.57 | 17.84/34.43 | 27.92/62.38 | 20.30/47.60 | – | – |
| | PromptGIP | 20.01/31.26 | 22.94/29.65 | 21.93/35.15 | 22.80/35.58 | 26.16/22.79 | 21.93/35.15 | 23.87/50.62 | 20.29/40.21 | – | – |
| | GenLV | 22.15/33.00 | 25.53/29.12 | 23.59/35.96 | 23.51/38.21 | 30.41/28.10 | 23.59/35.96 | 26.26/56.99 | 20.21/45.61 | – | – |
| Text-Prompt-based | PromptFix | 20.32/43.75 | 26.14/39.37 | 18.10/54.01 | 14.59/51.77 | 24.25/21.22 | 18.10/54.01 | 21.61/63.07 | 21.12/53.83 | 13.77/29.49 | – |
| | Pixwizard | 17.90/64.19 | 23.34/55.97 | 18.99/62.40 | 17.22/63.05 | 27.60/23.63 | 18.99/62.40 | 23.84/66.89 | 21.12/61.41 | 19.03/59.90 | – |
| Multi-Modal Instruction | OmniLV | 22.57/68.95 | 28.24/36.09 | 22.93/68.99 | 23.53/69.23 | 32.96/22.42 | 22.93/68.99 | 24.98/65.66 | 24.57/61.19 | 22.36/69.55 | 25.04/70.70 |

**Image Enhancement.** The enhancement dataset includes 14 major tasks with a total of 25 sub-tasks, covering tasks such as low-light correction, contrast enhancement, and saturation refinement. The dataset is composed of professionally edited reference images alongside algorithmically generated enhancement pairs, ensuring controlled transformations that align with perceptual quality.

**Weak-semantic Dense Prediction.** For dense prediction tasks, we compile annotated datasets for 10 tasks, including edge detection, depth estimation, and surface normal prediction. Each sample contains pixel-level ground truth annotations paired with descriptive task-specific instructions, facilitating multimodal learning.

**Image Stylization.** The stylization dataset spans 20 tasks, covering artistic transformations across various styles and techniques. It includes both real-world artistic works and style-transferred images generated by neural algorithms, ensuring a diverse range of stylish effects. We implement in-context learning on image stylization tasks due to the difficulty of defining task prompt.

**Dataset Summary and Test Set Construction.** OmniLV dataset comprises four major task categories with over 100 sub-tasks and approximately 40 million training instances. For publicly available datasets, we directly adopt their test sets for evaluation. For our synthesized tasks, we construct test sets based on DIV2K-val (Agustsson & Timofte, 2017), forming OmniLV-Test (OLV-T). OLV-T consists of 44 task-specific test sets, each containing 100 images, totaling 4,400 test images with 1K resolution. Further details on dataset partitioning and evaluation can be found in the Appendix A.

### 3.3 Model Training and Sampling Settings

The training of OmniLV is divided into three stages: In the first stage, we train the model with images at a resolution of $512^2$, focusing solely on single-image tasks. We use a constant learning rate of 1e-4 and train for 100k steps with a batch size of 512. The second stage adds in-context learning (ICL) tasks. We continue training for another 100k steps, maintaining the same learning rate of 1e-4 and batch size of 512. Finally, in the third stage, we increase the resolution to $1024^2$ and train on all tasks. The batch size is reduced to 128, and the learning rate remains at 1e-4. This final stage ensures that the model is trained to handle a variety of tasks and image sizes effectively. The model is trained using 16 A100 GPUs.

## 4 Experiments

### 4.1 Comparisons with Existing Works

As a universal model, OmniLV exhibits superior abilities for various low-level vision tasks, even compared with existing task-specific models. We compare our method with task-specific methods (X-Restormer (Chen et al., 2024d), MPRNet (Zamir et al., 2021), MAXIM (Tu et al., 2022), Diff-BIR (Lin et al., 2024b), GFPGAN (Wang et al., 2021), CodeFormer (Zhou et al., 2022), Retinexformer (Cai et al., 2023), MIRNet (Zamir et al., 2020), Depth Anything (D.A. ) (Yang et al., 2024a)), all-in-one methods (X-Restormer (Chen et al., 2024d), DA-CLIP (Luo et al., 2023), AutoDIR (Jiang et al., 2023)), visual prompt methods (PromptGIP (Liu et al., 2023a), GenLV (Chen et al., 2024e),

Table 6: Quantitative comparison on enhancement tasks.

| Category | Method | Brighten OLV-T(4 types) | Darken OLV-T(4 types) | Low light LOLv2-Real | Photoretouching MIT5K | Contrast Adjust OLV-T(4 types) | Saturation Adjust OLV-T(4 types) | Oversharpening OLV-T(1 type) |
|---|---|---|---|---|---|---|---|---|
| Specialized | Retinexformer | – | 16.72/65.73 | 22.79/59.30 | 16.12/63.24 | – | – | – |
| | MIRNet | – | 16.35/65.45 | 28.10/63.35 | 19.37/**65.59** | – | – | – |
| | MAXIM | – | 16.09/67.16 | **34.04/70.75** | 14.98/62.90 | – | – | – |
| All-in-One Restoration | DA-CLIP | – | 14.91/53.04 | 26.64/67.73 | – | – | – | – |
| | AutoDIR | – | 15.48/64.74 | 24.16/67.91 | – | – | – | – |
| Visual-Prompt-based | Painter | 12.00/36.77 | 13.96/37.09 | 29.44/53.82 | 17.19/58.39 | 12.55/35.99 | 13.25/36.66 | – |
| | PromptGIP | 15.46/35.43 | 17.85/33.89 | 21.35/38.18 | 16.57/43.02 | 15.80/33.75 | 16.63/34.49 | 16.63/38.74 |
| | GenLV | 21.11/40.16 | 21.70/39.31 | 21.01/50.84 | 24.91/56.08 | 21.58/39.46 | 20.87/40.29 | 21.69/37.08 |
| Text-Prompt-based | PromptFix | 10.55/57.78 | 10.15/54.40 | 17.16/63.54 | 11.09/52.89 | 11.34/57.76 | 12.31/58.49 | 14.93/56.01 |
| | PixWizard | 11.16/64.14 | 13.81/65.44 | 14.07/62.11 | 15.99/63.59 | 13.12/65.51 | 12.77/65.13 | 13.55/71.00 |
| Multi-Modal Instruction | OmniLV | 22.58/70.54 | 20.28/69.77 | 18.60/58.76 | 19.78/62.12 | 20.91/69.95 | 21.80/70.91 | 23.64/70.87 |

Table 7: Quantitative comparison on stylization tasks and weak-semantic dense prediction tasks.

(a) Stylization Tasks.

| Category | Method | LLF PSNR↑ | LLF FID↓ | PencilDrawing PSNR↑ | PencilDrawing FID↓ |
|---|---|---|---|---|---|
| Visual-Prompt-based | Painter | 13.64 | 120.3 | 8.434 | 157.5 |
| | PromptGIP | 22.87 | 50.61 | 21.35 | 132.5 |
| | GenLV | 25.66 | 29.53 | 28.29 | 38.70 |
| Multi-Modal Instruction | OmniLV | 21.72 | 24.16 | 20.33 | 54.18 |

(b) Dense Prediction Tasks.

| Method | Depth Esti. RMSE↓ | Method | Normal Esti. Mean Angle Error↓ | Method | HED MAE↓ |
|---|---|---|---|---|---|
| – | – | – | – | PromptGIP | 85.90 |
| D.A. | 0.291 | InvPT | 19.04 | GenLV | 44.28 |
| Pixwizard | 0.941 | Pixwizard | 19.65 | Pixwizard | 25.95 |
| OmniLV | 0.525 | OmniLV | 17.30 | OmniLV | 20.75 |

Painter (Wang et al., 2022)), and text-guided diffusion methods (PixWizard (Lin et al., 2024a), PromptFix (Yu et al., 2024b)). Some of them are constrained to generating images of fixed size. In our comparison, we resize the generated image to target image size to facilitate fair comparisons. We selected full-reference metric PSNR and no-reference metric MUSIQ (Ke et al., 2021) for quantitative comparison. The test sets include both our synthetic OLV-T and public datasets (RealBlur-J (Rim et al., 2020b), SIDD (Abdelhamed et al., 2018), Rain1400 (Fu et al., 2017), Snow100K-L (Liu et al., 2018a), DIV2K (Agustsson & Timofte, 2017), CelebA (Liu et al., 2018b), LOL-v2-Real (Yang et al., 2020), MIT5K (Bychkovsky et al., 2011b)). Detailed results are provided in Sec. D

**Image Restoration.** Tab. 5 demonstrates that OmniLV achieves the highest PSNR scores across all restoration benchmarks when compared with diffusion-based models. We demonstrate the qualitative comparisons in Fig. 7 on general low-level tasks. In addition, the MUSIQ scores of OmniLV are highly competitive on most benchmarks, further underscoring its strong performance. Notably, on the Blind Image Restoration (BIR) and Face benchmarks, OmniLV, as a universal model, attains performance levels that are comparable to those of state-of-the-art specialized models, thereby validating its effectiveness in handling diverse restoration tasks.

**Image Enhancement.** As reported in Tab. 6, OmniLV significantly improves upon existing enhancement methods. The improvements highlight OmniLV's ability to effectively enhance image quality while maintaining natural details and color fidelity.

**Dense Prediction.** Tab. 7b presents the performance of various models on dense prediction tasks, including depth estimation, normal estimation, and edge detection. Although OmniLV's performance still lags behind that of specialized models, it demonstrates significant improvements over baseline methods, underscoring its potential as a universal framework for dense-prediction tasks.

**Stylization.** Tab. 7a illustrates that OmniLV also performs well on stylization tasks such as Local Laplacian Filter (LLF) and Pencil Drawing (Lu et al., 2012). Since stylization tasks are challenging to describe using natural language, we employed visual prompts to guide the model in processing images. OmniLV obtains balanced results in terms of objective quality and perceptual quality, thus validating its versatility across diverse low-level vision tasks.

## 4.2 MORE EXPLORATION

**Text Prompt vs. Visual Prompt.** Our model supports both text prompt and visual prompt to guide the generation process. In Tab. 3, we present a detailed comparison between the two prompting methods across several low-level vision tasks, including deblurring, denoising, contrast adjustment, and saturation adjustment. Adopting both prompts can yield better quality scores.

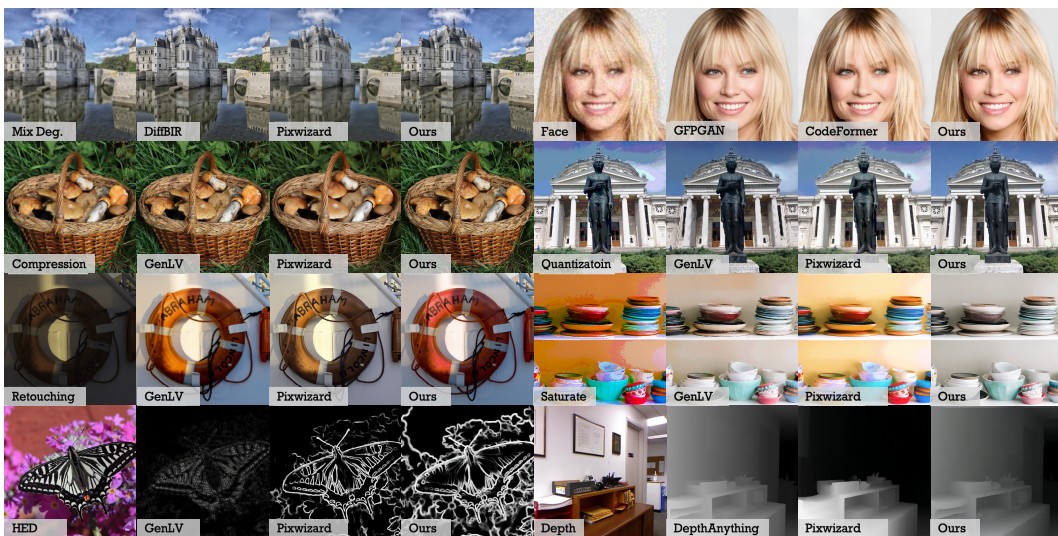

Figure 7: Comparison results for low-level vision tasks. More results can be found in the Supp.

**Relationship with High-Semantic Tasks.** We further investigate the relationship between low-level vision tasks and high-semantic tasks such as image generation or image editing tasks (Sheynin et al., 2024; Shi et al., 2020; Yildirim et al., 2023; Hui et al., 2024; Brooks et al., 2022; Zhang et al., 2023a; Zhao et al., 2024). As shown in Tab. 11, when high-semantic tasks are included in the training data, the performance on low-level vision tasks degrades. Specifically, performance for various tasks is consistently lower when high-semantic tasks are incorporated. This degradation arises because high-semantic tasks prioritize conceptual coherence and structural abstraction over pixel-accurate reconstruction, which conflicts with the objectives of low-level vision tasks that demand fine-grained texture recovery and precise detail preservation.

**Generalization Exploration.** We investigated the generalizability of OmniLV in terms of domain-specific adaptation and real-world robustness. Specifically, we selected images with various real-world degradations to comprehensively assess the model's performance. As shown in Fig. 8, OmniLV effectively restores images in these diverse conditions, demonstrating its robustness and versatility in handling complex real-world degradations, such as real-world restoration, deraining, desnowing, underwater image enhancement, and satellite image enhancement.

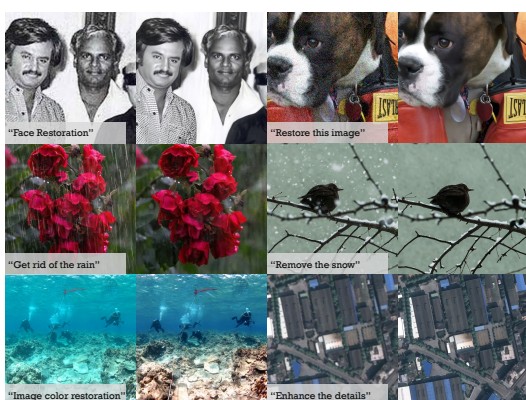

Figure 8: Examples of image restoration in various scenarios.

## 5 CONCLUSION

In this work, we introduced **OmniLV**, a unified multimodal framework for low-level vision that successfully handles over 100 sub-tasks, including image restoration, enhancement, weak-semantic dense prediction, and stylization. By leveraging both textual and visual prompts with generative priors, OmniLV demonstrates robust generalization, high-fidelity results, and flexibility across arbitrary resolutions. OmniLV achieves state-of-the-art performance in multiple low-level vision tasks and demonstrates promising generalization capabilities in real-world scenarios.

**Limitations.** Despite OmniLV's extensive capability on a wide range of low-level vision tasks, it does not always achieve optimal performance in certain specialized scenarios. Future work will focus on enhancing task-specific performance by refining model components and training strategies.

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

## A  TRAINING DATASET

Our dataset comprises four major types of low-level vision tasks: image restoration, image enhancement, weak-semantic dense prediction, and stylization. The dataset is constructed from both open-source datasets and internal synthesized data. Fig. 9 is a detailed version of the dataset composition:

- **Restoration:** Mix Degradation, Inpainting, SR (Agustsson & Timofte, 2017; Wang et al., 2023a), Rain, Outpainting, Face SR (Liu et al., 2018b; Karras et al., 2019), Ringing, Watermark (Liu et al., 2021a), Compression, Pixelate, Quantization, RL, Spatter, Face-Color, FaceInpait, Deshadow (Qu et al., 2017), NoiseGauss, Raindrop (Qian et al., 2018), Flare (Dai et al., 2023), Highlight (Dong et al., 2020), RealLLSR (Aakerberg et al., 2021), Reflection (Dong et al., 2021), CSD (Chen et al., 2021a), Snow100K (Liu et al., 2018a), SIDD (Abdelhamed et al., 2018), BlurGauss, BlurMotion, BlurGlass, BlurLens, BlurZoom, BlurJitter, NoiseSpeckle, NoiseSC, NoisePossion, NoiseImpulse, BSD (Martin et al., 2001), Rain13k (Jiang et al., 2020), GoPro (Nah et al., 2017), RealBlur (Rim et al., 2020a), Outdoor-Rain (Li et al., 2019), RESIDE (Li et al., 2018), RainDS (Quan et al., 2021), RealSnow Zhu et al. (2023), DenseHaze (Ancuti et al., 2019), NHHaze (Ancuti et al., 2020).

- **Enhancement:** Colorization, Brightening, Contrast Strengthening, Contrast Weakening, Darkening, Mosaic, Oversharpening, Saturation Strengthening, Saturation Weakening, Retouching (Bychkovsky et al., 2011a), Lowlight (Yang et al., 2020), Exposure Correction (Afifi et al., 2021), ISP (Ignatov et al., 2020b), Bokeh (Ignatov et al., 2020a), Vignet (Luo et al., 2024), White Balance (Afifi & Brown, 2020), Backlit (Liang et al., 2023).

- **Dense Prediction:** Depth Estimation, Normal Estimation, HED, Segmentation, Canny, Perceptual Edge, Hough Line, Saliency (Jiang et al., 2013).

- **Stylization:** Artist, LLF (Aubry et al., 2014), MultiTM, Pencil (Lu et al., 2012), PhotoRealistic, RTV (Xu et al., 2012).

For the synthesized portion, we generate corresponding input and output image pairs using various algorithms combined with our internally curated high-quality images. For the description prompts associated with the synthesized data, we provide annotations of varying lengths using BLIP (Li et al., 2022b), CogVLM (Wang et al., 2023b), and ShareGPT4V (Chen et al., 2024b). Additionally, we generate diverse task instructions for each task. In Fig. 11, we show examples of task prompts. System prompt used for generating task instructions is shown in Fig. 15.

## B  EVALUATION PROTOCOL

In our experiments, we use DIV2K-val as the source data and synthesize the corresponding test images with the same degradation algorithms applied in the training set. Since the output resolutions of the current baseline methods vary, we resize each output image to match the dimensions of the corresponding ground truth using `Bicubic` interpolation before computing evaluation metrics. PSNR

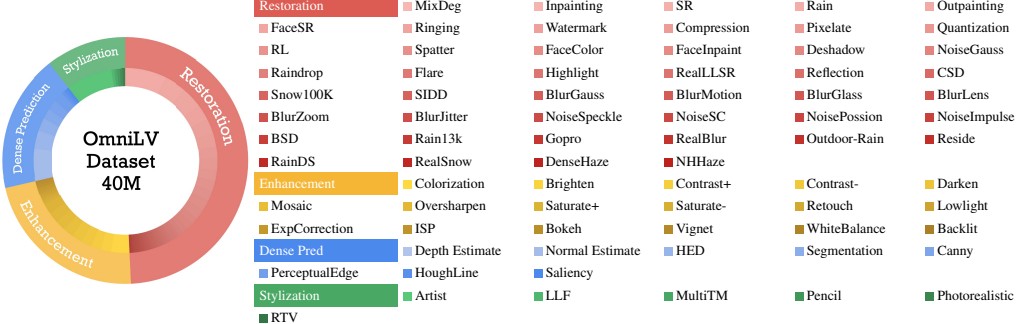

Figure 9: OmniLV dataset distribution with main categories.

and SSIM are calculated on the RGB color space. For depth estimation, we evaluate on the NYU-v2 test set (Silberman et al., 2012), which only provides metric depth. However, similar to Depth Anything, OmniLV predicts relative depth maps (disparity). Therefore, following the approach in (Ranftl et al., 2020), we convert the predicted disparity into metric depth for a fair comparison.

# C EXPERIMENT DETAILS

## C.1 STRUCTURE OF CONDITION ADAPTER

The condition adapter employs a 12 layer transformer with a linear layer to project condition features into DiT's latent space.

## C.2 ABLATION STUDY DETAILS

All ablation experiments are conducted under a consistent training configuration to ensure a fair comparison. Specifically, we adopt the first-stage training setup described in Section 3.3, using a resolution of $512^2$, 8 A100 GPUs, a batch size of 512, and a constant learning rate of 1e-4 for 100k training steps.

**Multimodal Encoding Variants.** To compare "separate versus unified" encoding strategies for integrating text instructions and visual exemplars, we use **Qwen-VL 2.5** as the unified multimodal encoder baseline. In the unified setting, both text and visual prompts are jointly encoded and passed to the diffusion model. In contrast, the separate encoding baseline decouples the two modalities, with text instructions processed by a language model and visual exemplars encoded via a visual VAE. Both variants are trained under identical conditions. The unified encoding model consistently underperforms due to modality interference, as discussed in the main paper and illustrated in Fig. 3. Following (Liu et al., 2021b; 2023b), we perform t-SNE analysis on dense prediction tasks for 200 data points each.

**Condition Integration Design.** We investigate five different strategies for integrating condition features into the diffusion model:

- **ControlNet-style injection**, where the condition is processed by a parallel branch and injected into the main model without updating the backbone.

- **Input Concatenation** directly concatenates the condition image with the input of the target image, and jointly feeds them into the model.

- **First-half Addition**, where condition features are added to the latent representations in the early layers.

- **Second-half Addition**, where addition occurs only in the later layers of the model.

- **Interleaved Addition**, where condition features are added in alternating layers throughout the network.

All variants use the same condition adapter described in Section C.1. As shown in Table 2, early integration (first half) consistently yields better performance, suggesting that early-stage guidance plays a critical role in conditioning effectiveness.

**In-Context Visual Prompting.** We evaluate two visual prompt integration paradigms: (1) **Input Concatenation**, where prompt tokens are directly concatenated to the input token sequence; and (2) **Projection-Addition**, where each visual prompt is projected to the latent space and added to the input latent. Both settings use the same projector architecture and number of visual exemplars. As shown in Table 1, projection-addition performs better in most tasks, which we attribute to better alignment and reduced representation conflict in the fused latent space.

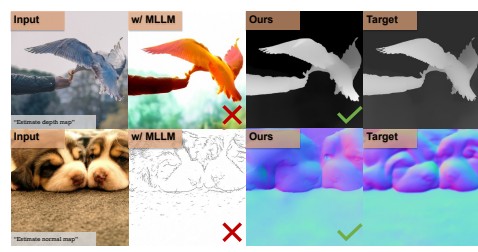

Figure 10: Task mismatch samples.

**Blur**

- Sharpen this blurred image as described
- Remove the blur from this photo based on the following instructions
- Enhance the clarity of this blurry image according to these details

**Compression**

- Restore this compressed image according to the description
- Improve the quality of this compressed photo based on these instructions
- Enhance the details of the compressed picture as specified

**Raining**

- Remove the rain from this image based on the following description
- Clear the raindrops from the photo according to these instructions
- Eliminate the rainy effect in the picture as described below:

**Haze**

- Remove the haze from this image
- Clear the hazy effect from the photo
- Eliminate the haze in the picture

**Inpainting**

- Inpaint this image with the following description
- Fill in the missing parts of the image based on these instructions
- Restore the damaged areas of the picture as described below

**Super Resolution**

- Enhance the resolution of this image
- Improve the image quality by increasing resolution
- Upscale the image resolution

**Spatter**

- Remove the spatter from this image as described
- Clean up the spatter marks in the photo according to these instructions
- Eliminate the spatter effect from the picture based on the following

**Quantization**

- Restore this quantized image according to the following details
- Improve the color depth of this quantized photo based on these instructions
- Enhance the color gradients in the quantized picture as specified

**Brighten**

- Adjust the brightness of this image to a normal level
- Reduce the excessive brightness of the photo
- Lower the overall luminosity of the picture to a standard range as specified

**Darken**

- Lighten this image based on the following details
- Increase the brightness of the photo according to these instructions
- Raise the overall luminosity of the picture as specified

**Contrast**

- Modify the contrast of this image to enhance details as described
- Tune the contrast settings of the image using this information
- Refine the contrast ratio in the photo to achieve a more natural appearance as per these guidelines:

**Saturate**

- Adjust the saturation of this image as described
- Change the saturation levels in the picture based on the following
- Change the color saturation of the picture as instructed

**Oversharpen**

- Inpaint this image with the following description
- Fill in the missing parts of the image based on these instructions
- Restore the damaged areas of the picture as described below

**Retouch**

- Enhance the resolution of this image
- Improve the image quality by increasing resolution
- Upscale the image resolution

**Exposure**

- Remove the spatter from this image as described
- Clean up the spatter marks in the photo according to these instructions
- Eliminate the spatter effect from the picture based on the following

**BacklitEnhance**

- Restore this quantized image according to the following details
- Improve the color depth of this quantized photo based on these instructions
- Enhance the color gradients in the quantized picture as specified

**Canny Edge**

- Inpaint this image with the following description
- Fill in the missing parts of the image based on these instructions
- Restore the damaged areas of the picture as described below

**Normal Estimation**

- Enhance the resolution of this image
- Improve the image quality by increasing resolution
- Upscale the image resolution

**Depth Estimation**

- Remove the spatter from this image as described
- Clean up the spatter marks in the photo according to these instructions
- Eliminate the spatter effect from the picture based on the following

**HED Edge**

- Restore this quantized image according to the following details
- Improve the color depth of this quantized photo based on these instructions
- Enhance the color gradients in the quantized picture as specified

Figure 11: Examples of prompts for different tasks.

## C.3 TRAINING LOSS

Specifically, let $(x, y) \sim q$ denote a pair of output and input images, respectively, and let $z \sim \mathcal{N}(0, I)$ be a noise sample. We define a target velocity field $u \colon [0, 1] \times \mathbb{R}^d \times \mathbb{R}^d \to \mathbb{R}^d$, which induces a flow $\phi \colon [0, 1] \times \mathbb{R}^d \times \mathbb{R}^d \to \mathbb{R}^d$, that continuously transforms the noise distribution into the output image distribution conditioned on the input. This transformation is governed by the ordinary differential equation (ODE)

$$\frac{\mathrm{d}}{\mathrm{d}t}\phi_t(x \mid y) = u_t\big(\phi_t(x \mid y) \mid y\big), \tag{3}$$

with the initial condition $\phi_0(x \mid y) = x$.

In flow-based models, a neural network is trained to approximate the conditional expectation $\bar{u}_t = \mathbb{E}[u_t \mid x_t, y]$, which represents an average over all plausible velocity fields at the state $x_t$ given the conditioning variable $y$. Accordingly, we optimize our model using the conditional flow matching (CFM) objective as described in (Liu et al., 2022)

$$\mathcal{L}_{\text{CFM}}(\theta) = \mathbb{E}_{t,q(x_1,y),p_t(x|x_1)}||u_\theta(t, x, y) - u_t(x|x_1)|| \tag{4}$$

where $t \sim \mathcal{U}[0, 1]$, $x_1$ and $y$ are sampled from the data distribution, and $x \sim p_t(x|x_1)$.

## C.4 PROMPT CONFLICT

To study how OmniLV behaves under inconsistent conditioning, we construct deliberately conflicting cases where the text instruction and the visual exemplars specify different tasks. As illustrated in Fig. 13, when both prompts are provided, the output consistently follows the text instruction rather than the visual exemplars. We do not intentionally create such conflicting pairs in our training data. We believe this behavior arises because the data is dominated by samples with text instructions. Consequently, the model has learned to treat the text instruction as the primary source of task specification in the presence of conflicts.

## C.5 CROSS-DOMAIN TASK

Following the definition in GenLV (Chen et al., 2024e), we categorize low-level vision tasks into five domains: restoration, enhancement, stylization, dense prediction, and natural images. The cross-domain capability in our setting refers to a model's ability to not only accomplish tasks within a single domain but also handle tasks from multiple domains simultaneously. To assess this ability, we construct composite tasks that involve multiple domains at once. As illustrated in Fig. 14, OmniLV demonstrates strong instruction-following behavior and consistently produces correct results across these mixed-domain scenarios.

## D MORE RESULTS

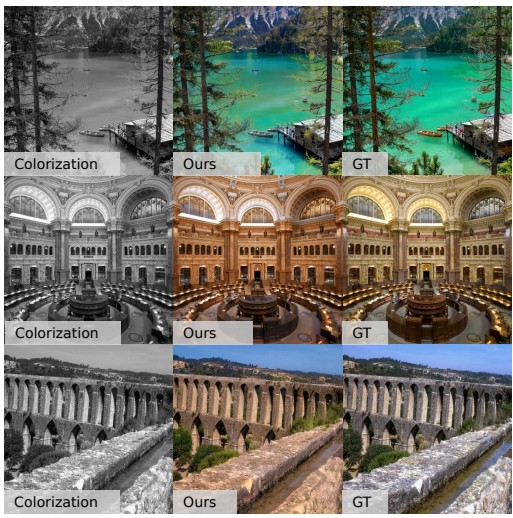

Figure 12: More results of colorization.

Table 8: Successful rates on some dense prediction tasks.

| Models | Image2Canny | Image2Depth | Image2HED |
|---|---|---|---|
| OmniGen | 80% | 75% | - |
| Pixwizard | 100% | 93% | 97% |
| OmniLV | 100% | 100% | 100% |

## D.1 DETAILED HUMAN EVALUATION PROTOCOL FOR OMNILV

**Study setup.** We use the Rapidata[1] platform to conduct pairwise human preference comparisons for evaluating low-level image restoration and enhancement quality. Each trial presents two model outputs (A/B) for the same degraded input, under a fixed unified instruction:

> Choose the better result. Focus on texture, fine details, overall image quality. If both are imperfect, pick the better one.

The A/B order is randomized per trial. Annotators are vetted beforehand and participate on a voluntary basis. We do not expose model identities to annotators.

**Datasets and models.** We evaluate three systems: our OmniLV, PixWizard, and PromptFix. All models operate on the same set of tasks. For each task, we construct a shared pool of 100 items. For each model pair, we sample 2,500 A/B comparisons across tasks (the same pool used for objective metrics), resulting in 7,500 total comparisons. Each comparison is judged by 5 independent annotators, yielding 37,500 votes in total.

**Preference aggregation and Elo computation.** For each model pair and item, Rapidata aggregates individual votes into a per-item preference count $(v_A, v_B)$ between models $A$ and $B$. We define the empirical win ratio for model $A$ as

$$s_A = \frac{v_A}{v_A + v_B}, \quad s_B = 1 - s_A. \tag{5}$$

To aggregate preferences across all items and all pairs into a single ranking, we adopt the robust Elo formulation. Given current Elo ratings $(R_A, R_B)$ for models $A$ and $B$, the expected win probability of $A$ is

$$E_A = \frac{1}{1 + 10^{\frac{R_B - R_A}{S}}}, \quad E_B = 1 - E_A, \tag{6}$$

where $S = 400$ is the scaling factor. We update the ratings using the continuous score $s_A$ instead of a binary win/loss signal:

$$
\begin{aligned}
R'_A &= \max\left(R_{\min}, \ R_A + K_{\text{eff}}(s_A - E_A)\right), \\
R'_B &= \max\left(R_{\min}, \ R_B + K_{\text{eff}}(s_B - E_B)\right),
\end{aligned}
\tag{7}
$$

where $K_{\text{eff}} = K \cdot \frac{v}{5}$ scales the base step size $K$ by the vote count $v = v_A + v_B$ (we use $v \approx 5$ in our setting), and $R_{\min}$ is a minimum floor rating.

**Robust aggregation.** To mitigate sensitivity to the order in which matches are processed, we shuffle the stream of pairwise comparisons and re-run the Elo updates for $T$ rounds. Denoting by $R_m^{(t)}$ the Elo rating of model $m$ after round $t$, we report the mean and standard deviation over rounds:

$$\bar{R}_m = \frac{1}{T} \sum_{t=1}^{T} R_m^{(t)}, \quad \sigma_m = \sqrt{\frac{1}{T} \sum_{t=1}^{T} \left(R_m^{(t)} - \bar{R}_m\right)^2}. \tag{8}$$

Elo gaps can be interpreted as win probabilities via the standard logistic mapping

$$P(A > B) = \frac{1}{1 + 10^{-\Delta R/S}}, \tag{9}$$

where $\Delta R = \bar{R}_A - \bar{R}_B$.

**Parameter setting.** Table 9 summarizes the Elo configuration used in our human evaluations.

**Results.** As summarized in Table 10, OmniLV achieves the highest human-preference Elo score, outperforming both PixWizard and PromptFix by substantial margins.

---

[1] https://www.rapidata.ai/

Table 9: Elo parameter setting for OmniLV human evaluation.

| Parameter | Value |
|---|---|
| Initial Elo rating | 1,000 |
| Elo scaling factor $S$ | 400 |
| Base K-factor $K$ | 24 |
| Minimum Elo rating $R_{\min}$ | 700 |
| Number of shuffle rounds $T$ | 50 |
| Votes per comparison $v$ | 5 |
| Models | 3 |
| Model pairs | 3 |
| Comparisons per pair | 2,500 |
| Total comparisons | 7,500 |
| Total votes | 37,500 |

Table 10: Human-preference Elo ratings on benchmarks. PromptFix is anchored at 1000 Elo..

| Model | Elo rating |
|---|---|
| OmniLV | 1,475 |
| PixWizard | 1,310 |
| PromptFix | 1,000 |

## D.2 DETAILED QUANTITATIVE RESULTS

In the main paper, we present quantitative results of several representative tasks. Here we provide a detailed quantitative results of more tasks, as summarized in Table 12, 13, 14, 15, 16, 17, 18, 19, 20, 21, 22, 23, 24. This section provides more results for diverse tasks. Fig. 12 presents the results of OmniLV on colorization. Fig. 16 presents more results of dense prediction, including Canny edge detection, HED, relative depth estimation, and normal estimation. Fig. 17 presents results of stylization achieved by in-context learning, mimicking local Laplacian filtering and pencil drawing. Fig. 18 and Fig. 19 present more results on image enhancement, including retouching, saturation adjustment, contrast adjustment, and mosaic removal. Fig. 20, Fig. 21, Fig. 22, Fig. 23, Fig. 24, and Fig. 25 present more results on image restoration, including face restoration, deblurring, deraining, dehazing, denoising, JPEG compression artifact removal, mixed degradation restoration, inpainting, deshadowing, and dewatermark. It can be seen that OmniLV consistently follows the text or visual prompt to conduct the various low-level vision tasks, while other methods often fail to follow the instruction and yield bad results.

## D.3 INSTRUCTION FOLLOWING ABILITY

Existing models often fail to adhere to the instructions and perform wrong transformations, especially for dense prediction tasks. In contrast, OmniLV shows strong instruction-following ability courtesy of the dedicated design choice described in Sec. 3.1.2. To illustrate it more clearly, we

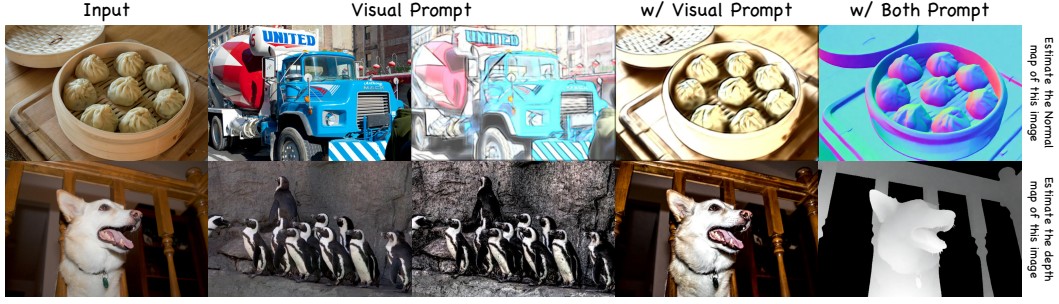

Figure 13: Results when providing conflict visual prompt and text instruction. Our model tends to follow text instruction when conflict happens.

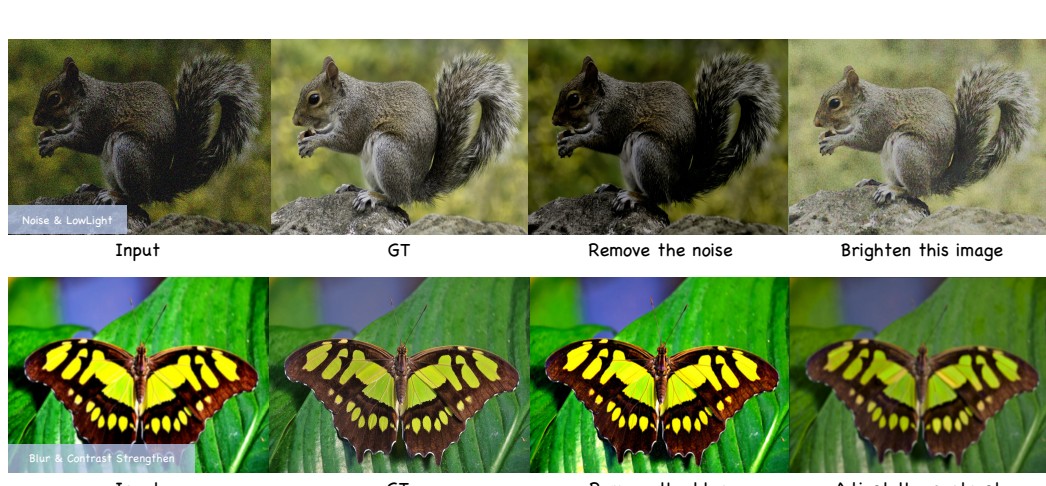

| Input | GT | Remove the noise | Brighten this image |

| Input | GT | Remove the blur | Adjust the contrast |

Figure 14: Results for cross-domain tasks. For cross-domain tasks, OmniLV can generate correct results with regard to the prompts.

### Prompt for OmniLV Instruction Generation

You are an instruction generator for low–level vision tasks.
Your goal is to produce clear, concise, task–specific instructions that describe exactly one low–level vision operation to be applied to an image.
For each task, you are given: 1. task category:A low–level vision task such as denoising, deblurring, deraining, dehazing, color enhancement, super–resolution, exposure correction, watermark removal, edge detection, etc. 2. Seed instructions: A set of mannually designed base instructions for the task.

Follow the rules strictly:
1.Task fidelity. The instruction must correspond precisely to the target task (e.g., deblurring, denoising, deraining, color correction, exposure adjustment, edge detection, style transfer, etc.). Do not introduce any unrelated high–level semantics (e.g., "make the image look like a painting of a cat" unless the task is stylization).
2.Instruction style
Use natural, user–friendly language.
Keep the instruction short: one sentence, typically 5—15 words.
Avoid technical jargon unless necessary (e.g., "canny edge" is allowed; "apply FFT–based deconvolution" is not).
3.Content requirements
Describe the transformation only.
Do not hallucinate content, modify objects, or add/remove semantics.
The instruction must not require external knowledge.
4.Diversity
Rephrase meaningfully without changing the underlying task.
5.Output format
Output only the generated instruction sentence, without additional explanations.

Example behaviors:
For deblurring: "Remove the motion blur from this image."
For denoising: "Reduce the noise and clean up the image."
For contrast adjustment: "Increase the contrast to make details stand out."
For depth estimation: "Estimate a depth map for this image."

Follow the guidelines above to generate high–quality, task–aligned instructions.

Figure 15: System Prompt for instruction generation.

Table 11: Ablation study for the training data.

| Training Data | Blur | | Noise | | Quantization | | Rain | |
|---|---|---|---|---|---|---|---|---|
| | PSNR↑ | MUSIQ↑ | PSNR↑ | MUSIQ↑ | PSNR↑ | MUSIQ↑ | PSNR↑ | MUSIQ↑ |
| Restoration | 21.55 | 54.77 | 21.49 | 55.54 | 20.05 | 55.93 | **20.86** | 56.05 |
| Restoration & Enhancement | **21.71** | **55.05** | **21.59** | **55.57** | **20.67** | **56.34** | 20.51 | **56.73** |

compute the success rates of OmniLV on some dense prediction tasks compared to OmniGen (Xiao et al., 2024) and PixWizard (Lin et al., 2024a). We deem a case as successful when the model performs the intended task, no matter how well. Tab. 8 presents the results, showing the superior task alignment ability of OmniLV.

## D.4 EFFECTIVENESS OF MULTI-TASK TRAINING

OmniLV, as a universal model for over 100 low-level vision tasks, can potentially learn shared knowledge across tasks and thus perform better in specific tasks. We ablate training data to investigate this. Tab. 11 shows that adding enhancement tasks to restoration improves performance due to their shared task characteristics. This demonstrates the benefits of multi-task training, which is a unique strength of OmniLV.

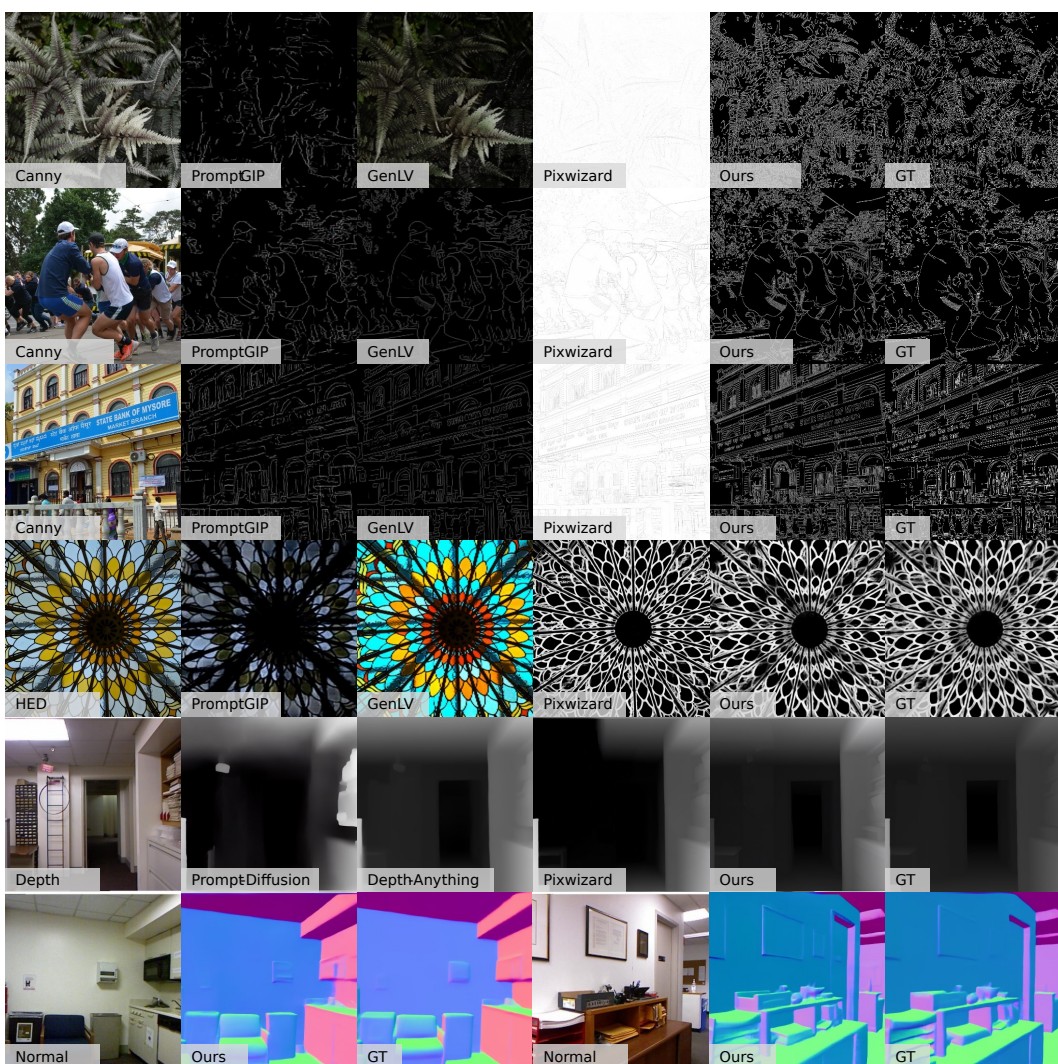

Figure 16: More results of dense prediction.

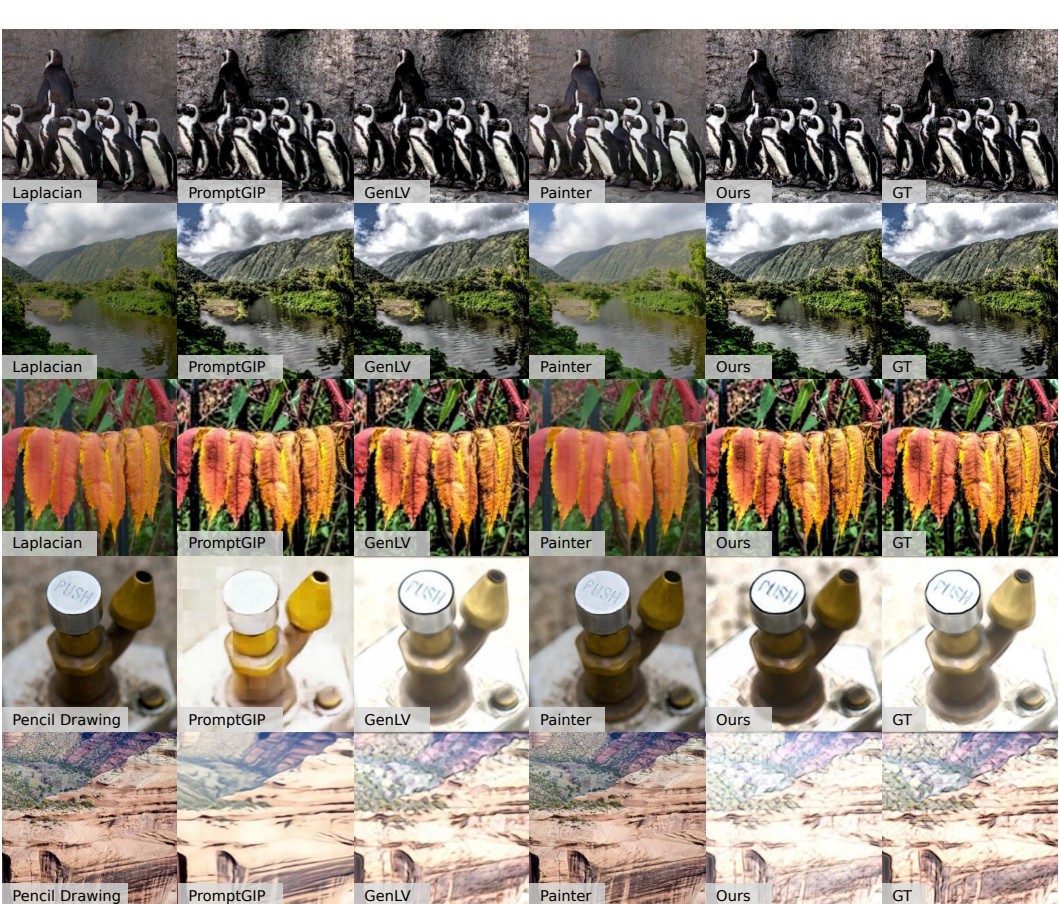

Figure 17: More results of stylization.

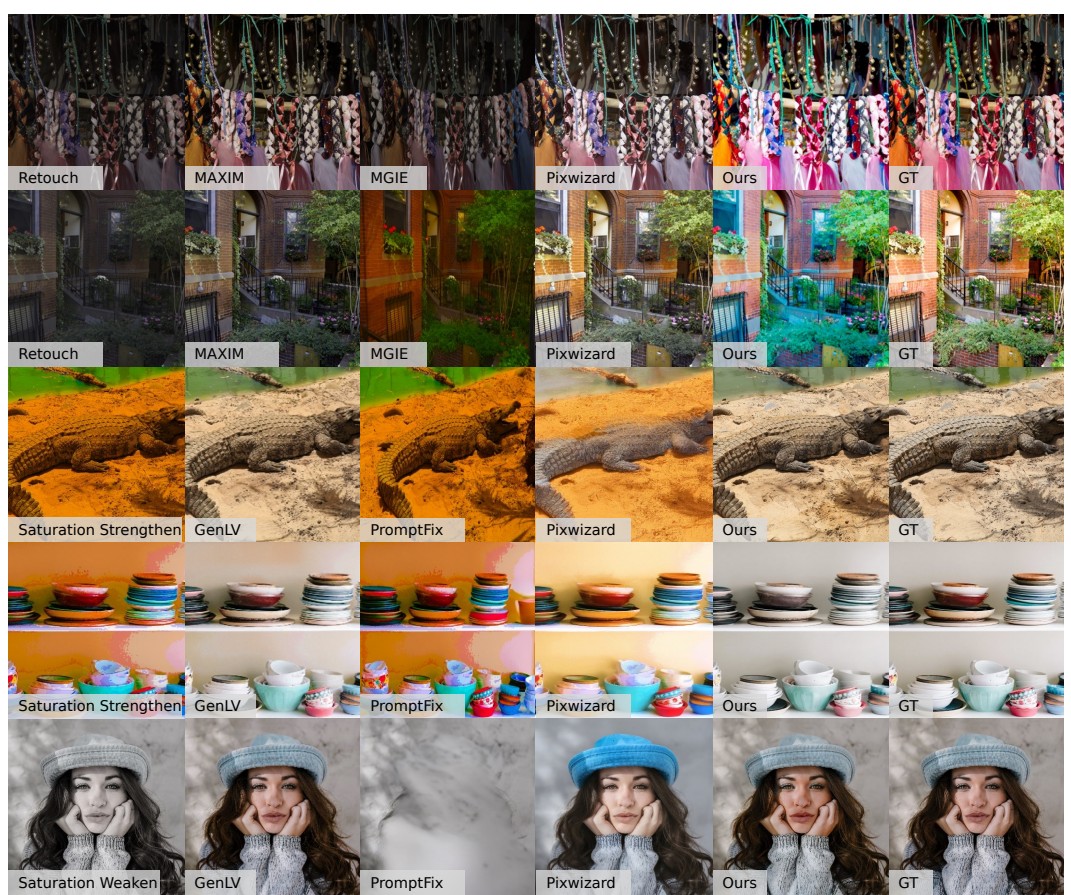

Figure 18: More results of image enhancement.

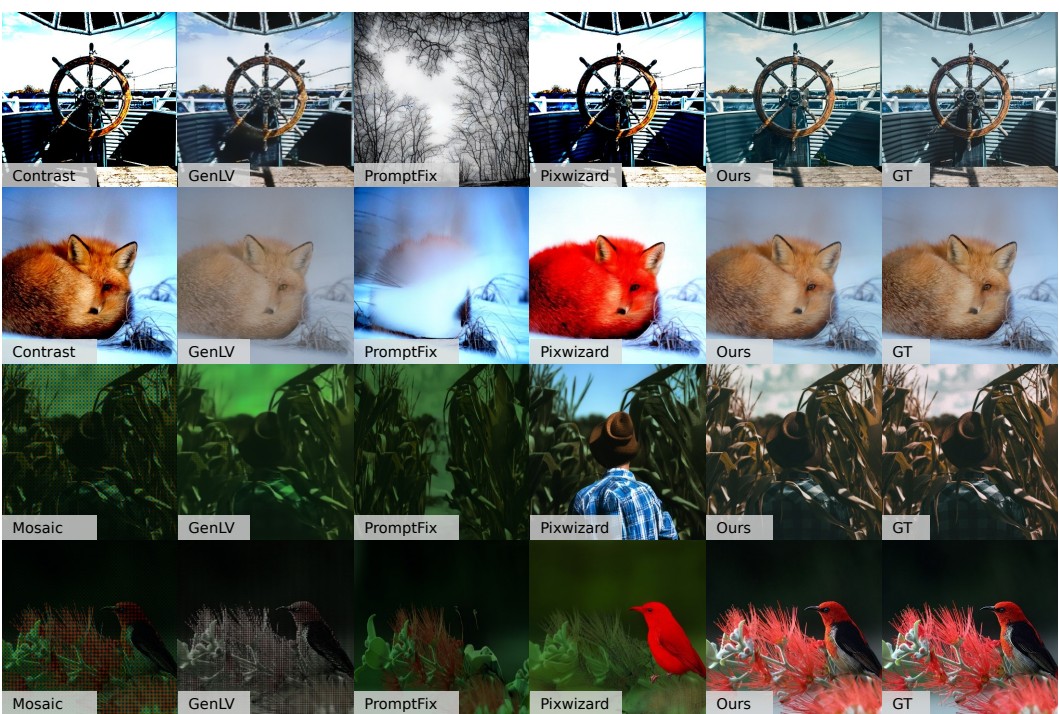

Figure 19: More results of image enhancement.

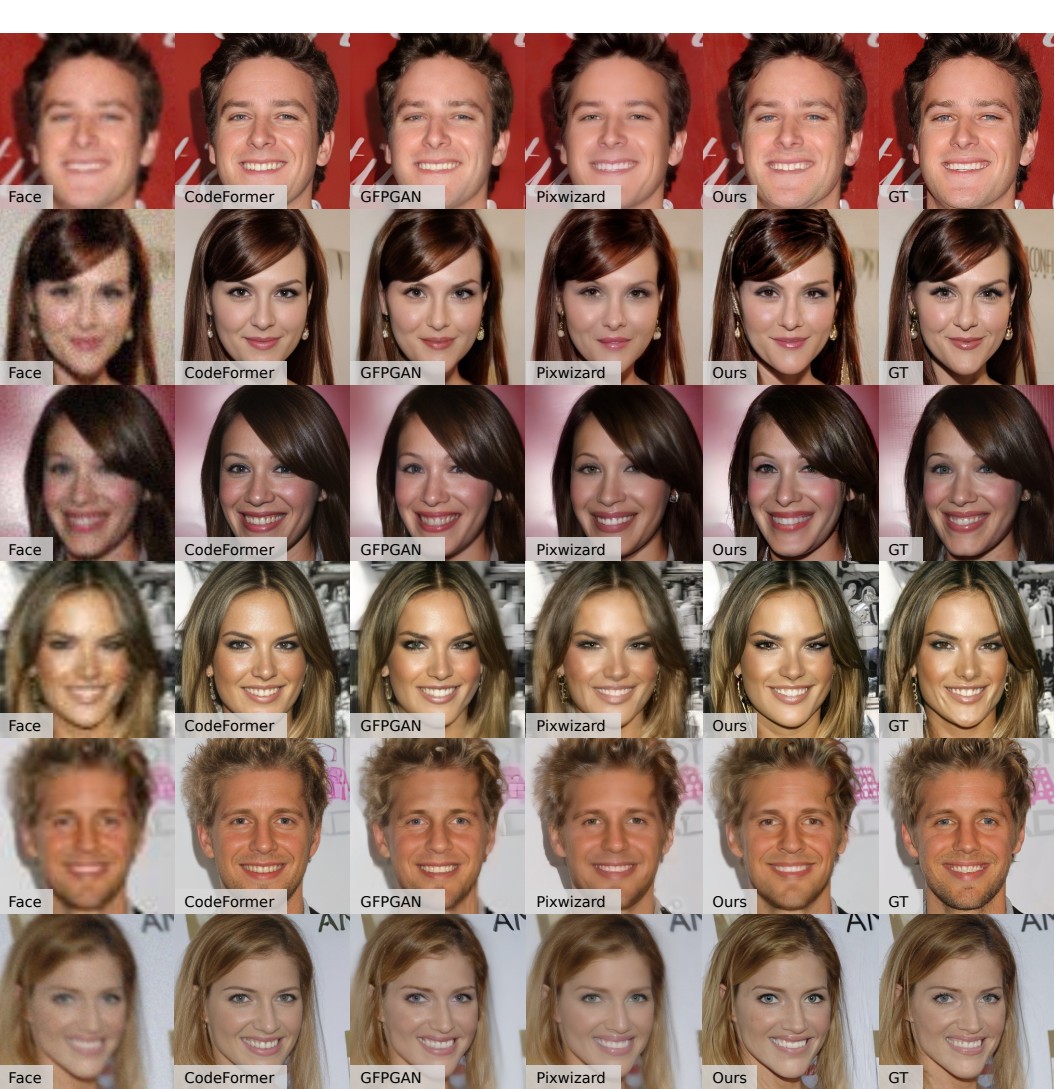

Figure 20: More results of face restoration.

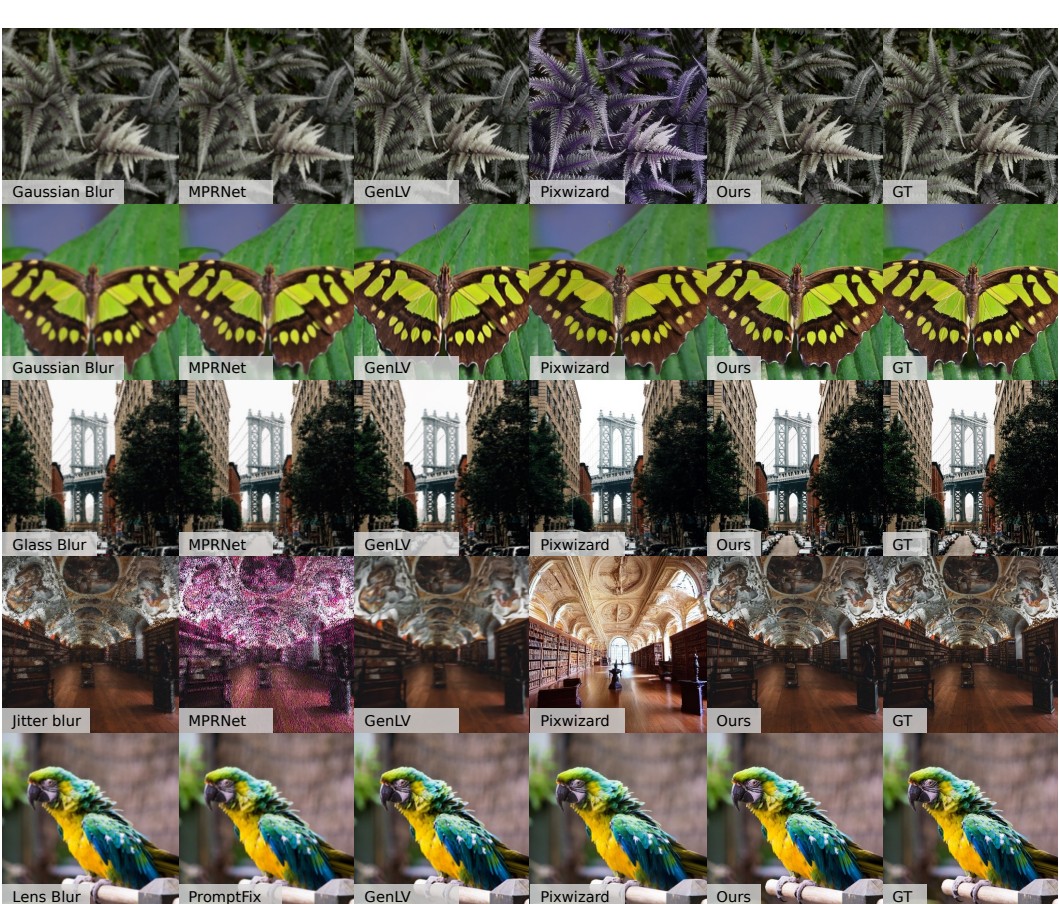

Figure 21: More results of deblurring.

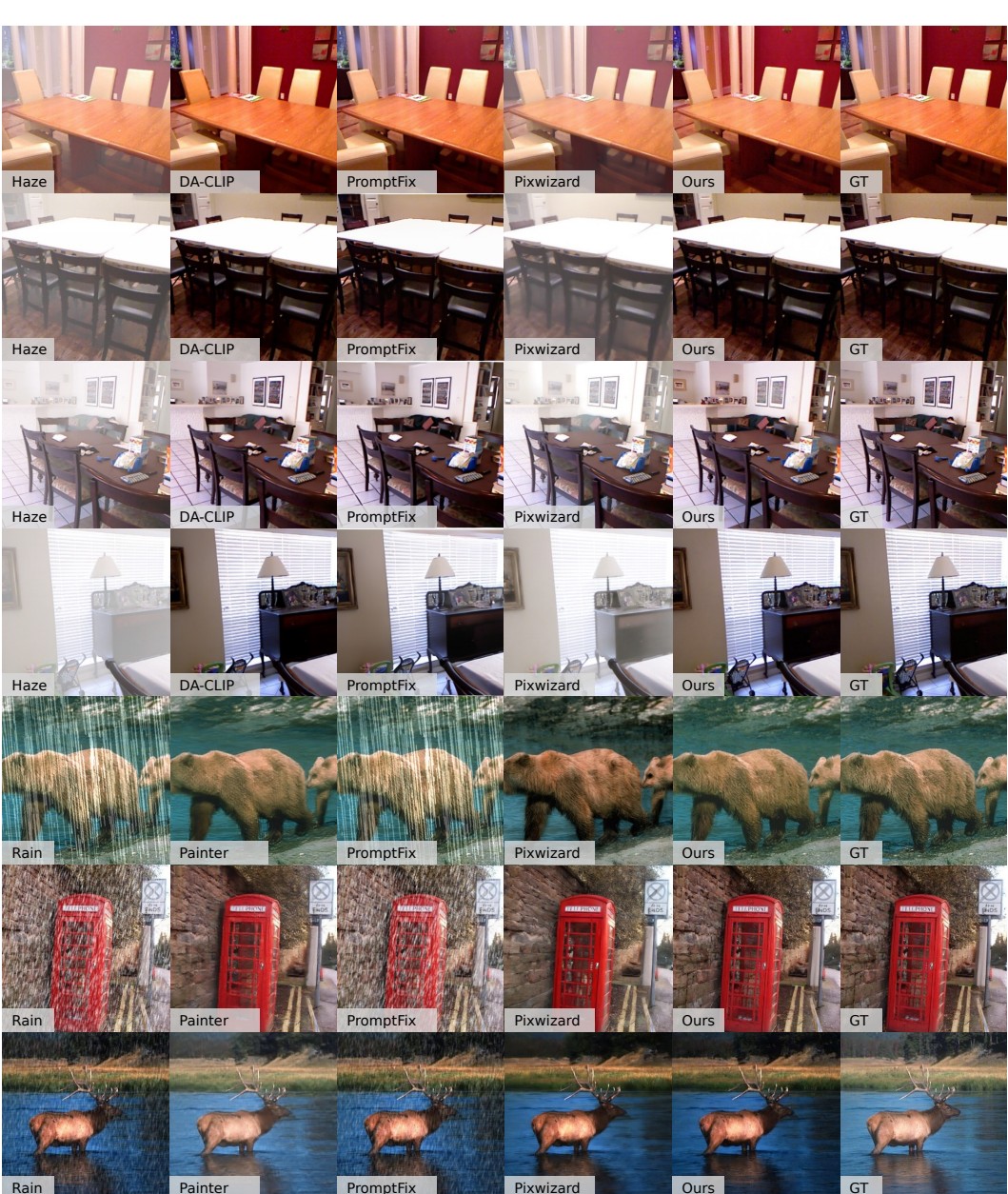

Figure 22: More results of dehazing and deraining.

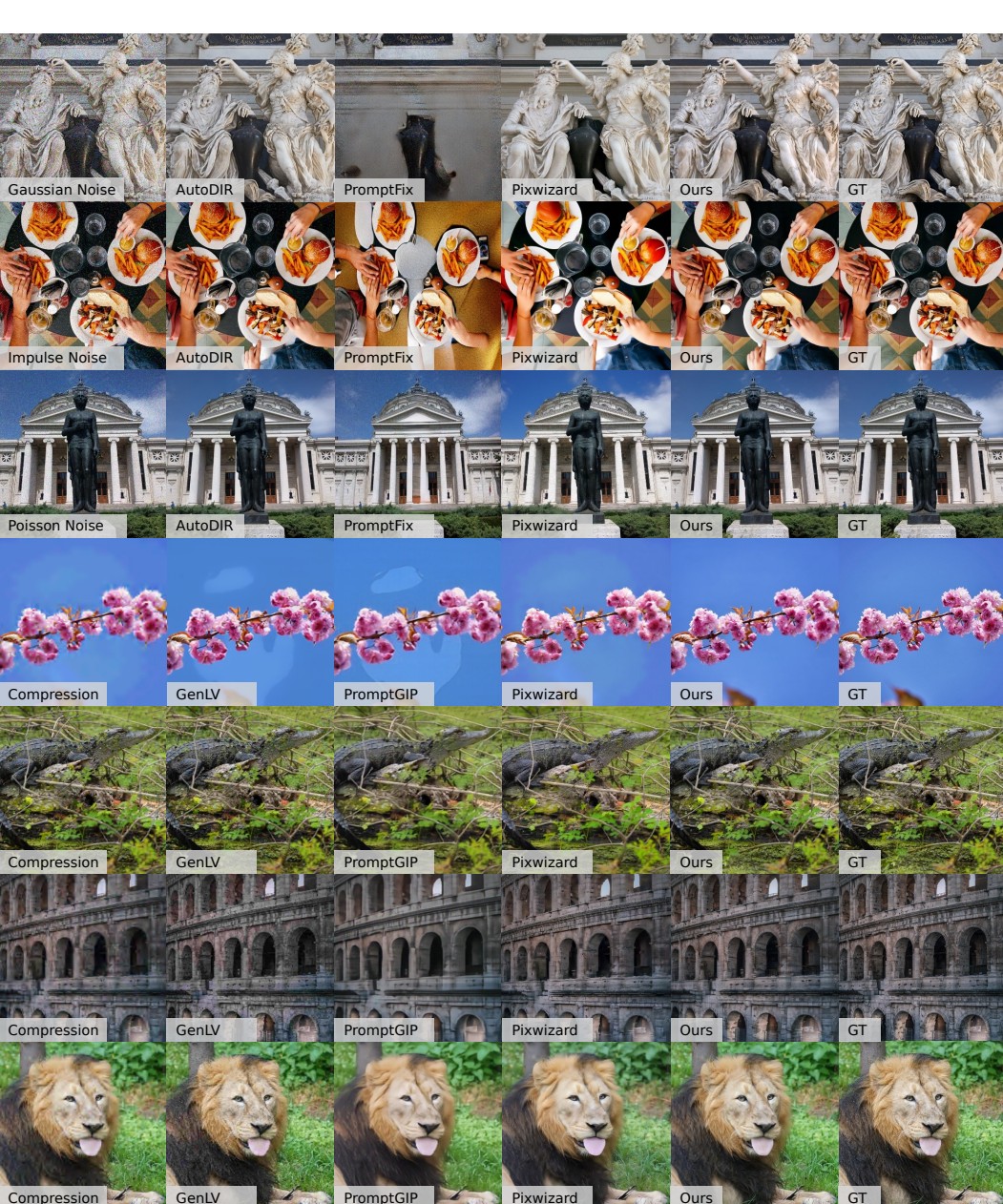

Figure 23: More results of denoising and compression artifact removal.

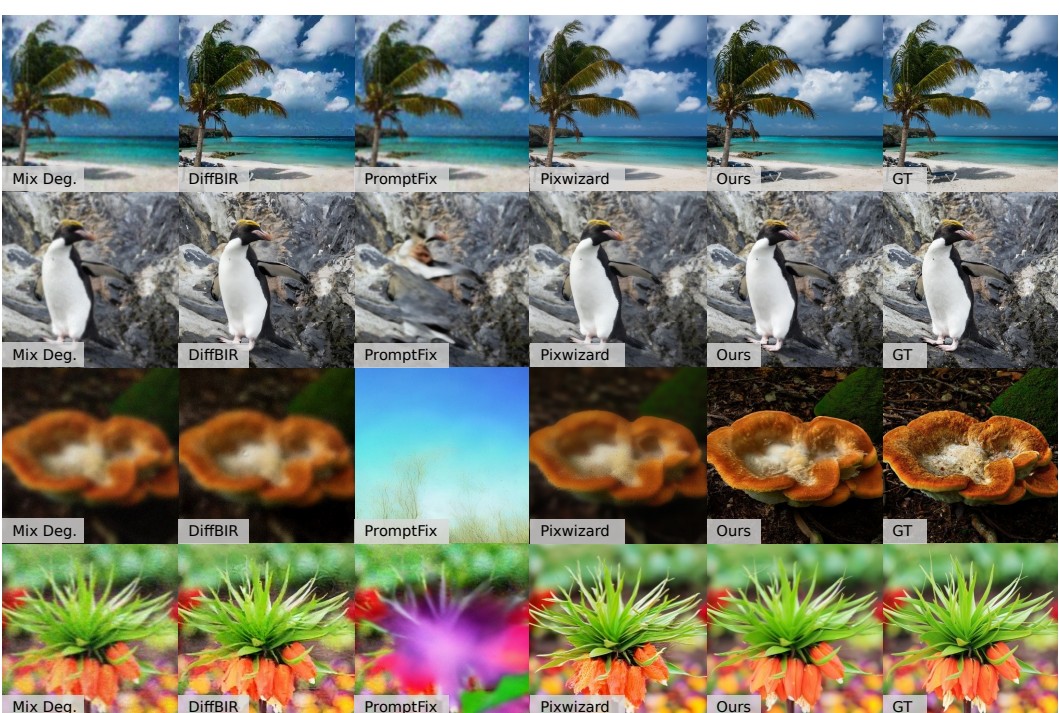

Figure 24: More results of mixed degradation restoration.

| Category | Method | Blur (Gaussian) | | | | Blur (Glass) | | | |
|----------|--------|-------|-------|-------|--------|-------|-------|-------|--------|
| | | PSNR↑ | SSIM↑ | FID↓ | MUSIQ↑ | PSNR↑ | SSIM↑ | FID↓ | MUSIQ↑ |
| Specialized | X-Restormer | 23.34 | 0.6375 | 63.75 | 27.99 | 22.40 | 0.6187 | 80.10 | 28.71 |
| | MPRNet | 23.28 | 0.6309 | 69.91 | 27.14 | 21.65 | 0.5884 | 80.55 | 30.58 |
| | MAXIM | 23.32 | 0.6353 | 70.24 | 26.33 | 22.26 | 0.6102 | 77.62 | 29.52 |
| All-in-One | X-Restormer | 23.50 | 0.6429 | 65.13 | 27.86 | 21.16 | 0.5828 | 80.40 | 28.17 |
| | DA-CLIP | 19.39 | 0.5306 | 75.38 | 32.24 | 21.02 | 0.5837 | 83.38 | 29.22 |
| | AutoDIR | 24.01 | 0.6712 | 43.38 | 46.35 | 19.28 | 0.5467 | 79.28 | 35.70 |
| | GenLV | 24.11 | 0.6652 | 51.40 | 32.99 | 22.32 | 0.6172 | 73.08 | 30.96 |
| Visual Prompt | PromptGIP | 21.10 | 0.5552 | 128.60 | 31.04 | 20.81 | 0.5523 | 147.40 | 31.93 |
| | Painter | 16.84 | 0.4638 | 166.90 | 25.03 | 16.80 | 0.4808 | 166.80 | 27.40 |
| | Prompt-Diffusion | 9.339 | 0.2591 | 174.5 | 49.93 | 9.389 | 0.2406 | 168.4 | 56.39 |
| Text Prompt | Instruct-Pix2Pix | 16.22 | 0.4955 | 127.60 | 34.46 | 16.04 | 0.4778 | 119.90 | 37.34 |
| | MGIE | 17.59 | 0.5004 | 110.00 | 27.12 | 16.23 | 0.4424 | 134.30 | 32.89 |
| | PromptFix | 24.48 | 0.7217 | 78.13 | 33.05 | 22.22 | 0.6649 | 99.78 | 40.05 |
| | PixWizard | 20.49 | 0.5367 | 59.23 | 67.66 | 19.65 | 0.5162 | 59.58 | 65.54 |
| Multi-Modal | OmniLV | 23.29 | 0.6437 | 18.19 | 67.98 | 22.41 | 0.6299 | 25.43 | 68.45 |

Table 12: Restoration on blur corruption (Gaussian and glass). Higher is better for PSNR/SSIM/MUSIQ, lower is better for FID.

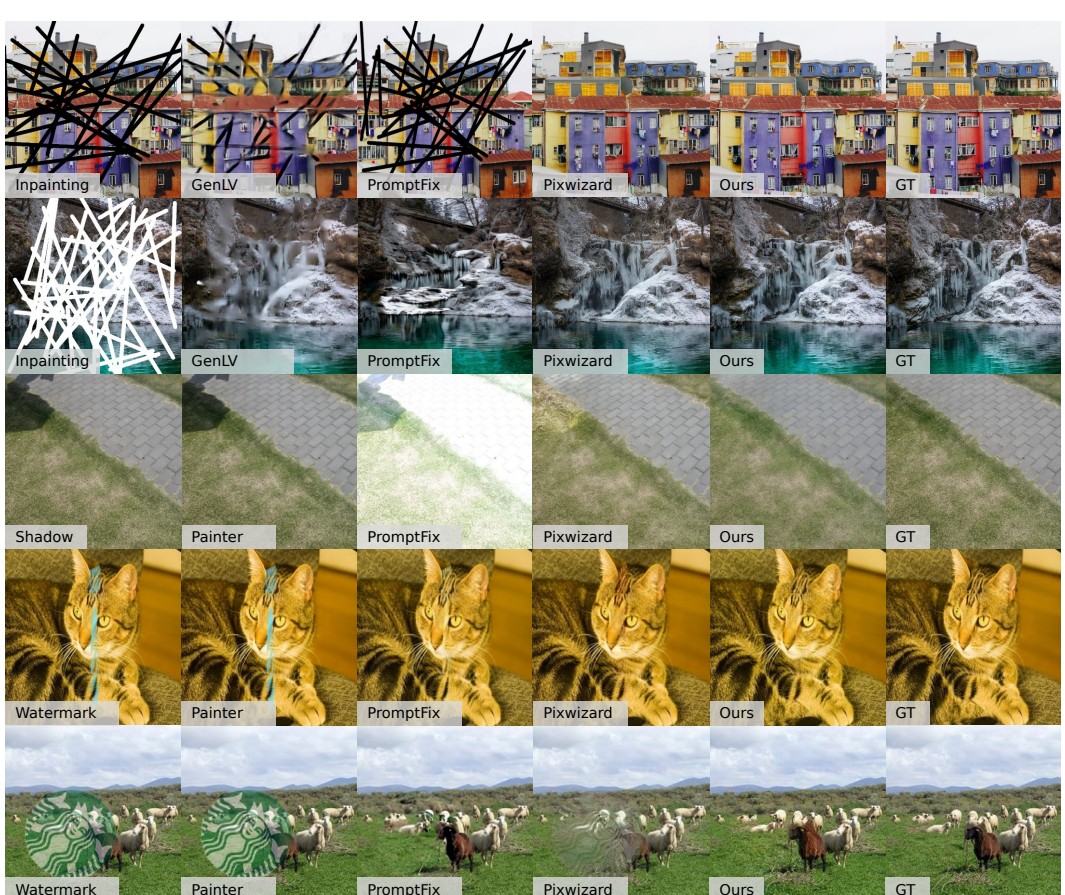

Figure 25: More results of image restoration.

| Category | Method | Blur (Motion) | | | | Compression (JPEG) | | | |
|---|---|---|---|---|---|---|---|---|---|
| | | PSNR↑ | SSIM↑ | FID↓ | MUSIQ↑ | PSNR↑ | SSIM↑ | FID↓ | MUSIQ↑ |
| Specialized | X-Restormer | 19.42 | 0.5485 | 28.96 | 61.81 | – | – | – | – |
| | MPRNet | 19.73 | 0.5673 | 28.46 | 59.06 | – | – | – | – |
| | MAXIM | 19.70 | 0.5634 | 30.38 | 58.40 | – | – | – | – |
| All-in-One | X-Restormer | 20.30 | 0.5961 | 50.25 | 42.80 | – | – | – | – |
| | DA-CLIP | 19.63 | 0.5665 | 64.25 | 36.67 | – | – | – | – |
| | AutoDIR | 18.77 | 0.5452 | 37.98 | 47.88 | – | – | – | – |
| Visual Prompt | GenLV | 20.72 | 0.5897 | 64.54 | 31.81 | 23.85 | 0.6861 | 66.70 | 36.43 |
| | PromptGIP | 19.01 | 0.5184 | 165.70 | 30.52 | 22.33 | 0.6096 | 93.84 | 36.89 |
| | Painter | 16.53 | 0.4668 | 138.30 | 29.53 | 20.93 | 0.6405 | 97.07 | 38.40 |
| | Prompt-Diffusion | 9.446 | 0.2502 | 164.10 | 56.75 | 9.710 | 0.2433 | 154.50 | 62.26 |
| Text Prompt | Instruct-Pix2Pix | 15.88 | 0.4658 | 112.30 | 37.93 | 16.53 | 0.4864 | 97.80 | 46.43 |
| | MGIE | 15.06 | 0.4297 | 111.60 | 36.79 | 16.39 | 0.4912 | 118.10 | 51.46 |
| | PromptFix | 20.00 | 0.6149 | 89.82 | 46.47 | 15.96 | 0.5802 | 187.10 | 58.75 |
| | PixWizard | 17.42 | 0.4763 | 64.04 | 65.15 | 18.81 | 0.5300 | 63.87 | 60.02 |
| Multi-Modal | OmniLV | 23.36 | 0.6697 | 16.73 | 69.40 | 23.79 | 0.6854 | 29.05 | 69.12 |

Table 13: Restoration on motion blur and JPEG compression. Higher is better for PSNR/SSIM/MUSIQ, lower is better for FID.

| Category | Method | Noise (Gaussian) | | | | Noise (Poisson) | | | |
|---|---|---|---|---|---|---|---|---|---|
| | | PSNR↑ | SSIM↑ | FID↓ | MUSIQ↑ | PSNR↑ | SSIM↑ | FID↓ | MUSIQ↑ |
| Specialized | X-Restormer | 27.37 | 0.7856 | 40.85 | 68.27 | **29.52** | **0.8603** | 45.46 | 69.24 |
| | MPRNet | 23.67 | 0.6909 | 65.67 | 43.58 | 25.81 | 0.7714 | 40.49 | 49.86 |
| | MAXIM | 22.88 | 0.6959 | 68.16 | 46.89 | 26.35 | 0.7938 | 44.97 | 54.66 |
| All-in-One | X-Restormer | 27.51 | 0.8022 | 23.94 | 69.65 | 28.63 | 0.8232 | 16.94 | 69.01 |
| | DA-CLIP | 23.16 | 0.5769 | 70.62 | 45.54 | 23.19 | 0.5821 | 61.60 | 43.08 |
| | AutoDIR | 26.78 | 0.7882 | 20.34 | 60.15 | 28.23 | 0.8559 | 12.61 | 63.12 |
| Visual Prompt | GenLV | 23.10 | 0.6406 | 79.11 | 38.48 | 23.91 | 0.6938 | 60.56 | 39.65 |
| | PromptGIP | 22.54 | 0.6046 | 84.05 | 35.32 | 22.98 | 0.6389 | 72.32 | 35.69 |
| | Painter | 17.56 | 0.5603 | 123.50 | 36.22 | 16.96 | 0.5531 | 132.70 | 38.34 |
| | Prompt-Diffusion | 9.779 | 0.2199 | 160.30 | 60.41 | 9.790 | 0.2383 | 154.20 | 62.00 |
| | Instruct-Pix2Pix | 14.70 | 0.3935 | 114.20 | 46.80 | 15.28 | 0.4316 | 101.10 | 48.50 |
| Text Prompt | MGIE | 14.47 | 0.2379 | 121.00 | 40.99 | 15.70 | 0.3078 | 92.45 | 48.28 |
| | PromptFix | 13.99 | 0.4334 | 207.2 | 50.08 | 15.16 | 0.5230 | 185.8 | 55.00 |
| | PixWizard | 17.05 | 0.4340 | 74.85 | 62.13 | 15.80 | 0.4423 | 76.09 | 62.52 |
| Multi-Modal | OmniLV | 23.21 | 0.6405 | 26.17 | 69.78 | 23.98 | 0.6778 | 19.30 | 69.82 |

Table 14: Restoration on additive noise corruption (Gaussian and Poisson). Higher is better for PSNR/SSIM/MUSIQ, lower is better for FID.

| Category | Method | Pixelate | | | | Quantization (Hist) | | | |
|---|---|---|---|---|---|---|---|---|---|
| | | PSNR↑ | SSIM↑ | FID↓ | MUSIQ↑ | PSNR↑ | SSIM↑ | FID↓ | MUSIQ↑ |
| Specialized | X-Restormer | – | – | – | – | – | – | – | – |
| | MPRNet | – | – | – | – | – | – | – | – |
| | MAXIM | – | – | – | – | – | – | – | – |
| All-in-One | X-Restormer | – | – | – | – | – | – | – | – |
| | DA-CLIP | – | – | – | – | – | – | – | – |
| | AutoDIR | – | – | – | – | – | – | – | – |
| Visual Prompt | GenLV | 24.28 | 0.6977 | 33.97 | 36.89 | 21.83 | 0.6701 | 66.33 | 37.99 |
| | PromptGIP | 22.01 | 0.6025 | 92.48 | 34.61 | 15.10 | 0.5046 | 159.40 | 35.37 |
| | Painter | 18.43 | 0.5530 | 110.60 | 36.02 | 12.50 | 0.4319 | 186.60 | 34.92 |
| | Prompt-Diffusion | 9.689 | 0.2587 | 153.40 | 61.77 | 9.180 | 0.2455 | 155.70 | 63.33 |
| | Instruct-Pix2Pix | 15.69 | 0.4556 | 90.60 | 56.01 | 14.15 | 0.4504 | 98.80 | 47.22 |
| Text Prompt | MGIE | 13.90 | 0.3517 | 172.70 | 49.02 | 12.90 | 0.3921 | 118.30 | 58.23 |
| | PromptFix | 18.34 | 0.6494 | 143.60 | 59.09 | 12.51 | 0.5430 | 193.60 | 59.29 |
| | PixWizard | 14.31 | 0.4422 | 103.40 | 57.60 | 13.98 | 0.4560 | 86.58 | 65.38 |
| Multi-Modal | OmniLV | 23.45 | 0.6673 | 16.70 | 68.50 | 20.46 | 0.6457 | 33.85 | 69.46 |

Table 15: Restoration on pixelation and histogram quantization corruption. Higher is better for PSNR/SSIM/MUSIQ, lower is better for FID.

| Category | Method | Quantization (Median) | | | | Quantization (Otsu) | | | |
|---|---|---|---|---|---|---|---|---|---|
| | | PSNR↑ | SSIM↑ | FID↓ | MUSIQ↑ | PSNR↑ | SSIM↑ | FID↓ | MUSIQ↑ |
| Specialized | X-Restormer | – | – | – | – | – | – | – | – |
| | MPRNet | – | – | – | – | – | – | – | – |
| | MAXIM | – | – | – | – | – | – | – | – |
| All-in-One | X-Restormer | – | – | – | – | – | – | – | – |
| | DA-CLIP | – | – | – | – | – | – | – | – |
| | AutoDIR | – | – | – | – | – | – | – | – |
| Visual Prompt | GenLV | 22.19 | 0.6725 | 84.79 | 35.59 | 20.83 | 0.6684 | 69.59 | 38.31 |
| | PromptGIP | 21.25 | 0.5963 | 116.40 | 34.35 | 17.87 | 0.5464 | 129.50 | 35.73 |
| | Painter | 16.31 | 0.5247 | 153.00 | 36.33 | 14.29 | 0.4893 | 174.70 | 36.73 |
| | Prompt-Diffusion | 9.508 | 0.2554 | 152.30 | 63.32 | 9.399 | 0.2488 | 150.90 | 63.76 |
| | Instruct-Pix2Pix | 16.94 | 0.5018 | 91.10 | 49.15 | 15.37 | 0.4445 | 96.42 | 48.65 |
| Text Prompt | MGIE | 14.88 | 0.4414 | 98.60 | 59.51 | 14.30 | 0.4157 | 109.30 | 58.08 |
| | PromptFix | 16.01 | 0.6160 | 182.10 | 59.56 | 14.36 | 0.5616 | 172.80 | 61.06 |
| | PixWizard | 15.39 | 0.4769 | 76.05 | 65.62 | 15.20 | 0.4637 | 78.45 | 65.69 |
| Multi-Modal | OmniLV | 22.26 | 0.6758 | 32.54 | 69.68 | 19.75 | 0.6428 | 35.88 | 69.35 |

Table 16: Restoration on histogram quantization corruption (Median/Otsu variants). Higher is better for PSNR/SSIM/MUSIQ, lower is better for FID.

| Category | Method | Rain | | | | Ringing | | | |
|---|---|---|---|---|---|---|---|---|---|
| | | PSNR↑ | SSIM↑ | FID↓ | MUSIQ↑ | PSNR↑ | SSIM↑ | FID↓ | MUSIQ↑ |
| Specialized | X-Restormer | **27.10** | **0.8691** | 45.87 | **71.34** | – | – | – | – |
| | MPRNet | 25.16 | 0.8397 | 64.83 | 69.40 | – | – | – | – |
| | MAXIM | 25.82 | 0.8537 | 55.18 | 71.15 | – | – | – | – |
| All-in-One | X-Restormer | 23.28 | 0.7832 | 91.22 | 69.00 | – | – | – | – |
| | DA-CLIP | 23.15 | 0.7231 | 58.17 | 53.18 | – | – | – | – |
| | AutoDIR | 25.33 | 0.8104 | 42.63 | 64.59 | – | – | – | – |
| Visual Prompt | GenLV | 21.05 | 0.6146 | 107.10 | 36.43 | 25.10 | 0.7266 | 34.04 | 39.28 |
| | PromptGIP | 21.17 | 0.5868 | 100.50 | 34.25 | 23.34 | 0.6488 | 60.82 | 35.76 |
| | Painter | 22.48 | 0.6649 | 74.64 | 41.30 | 16.16 | 0.5224 | 159.40 | 37.51 |
| | Prompt-Diffusion | 9.743 | 0.1959 | 205.70 | 58.65 | 9.735 | 0.2703 | 147.90 | 63.23 |
| Text Prompt | Instruct-Pix2Pix | 14.07 | 0.3412 | 173.60 | 42.28 | 16.76 | 0.5076 | 81.44 | 49.66 |
| | MGIE | 14.15 | 0.3252 | 200.60 | 54.41 | 15.96 | 0.4654 | 122.40 | 53.27 |
| | PromptFix | 11.69 | 0.3404 | 246.60 | 57.63 | 18.22 | 0.6788 | 161.60 | 61.62 |
| | PixWizard | 17.61 | 0.5003 | 64.05 | 70.66 | 21.55 | 0.6188 | 35.08 | 65.14 |
| Multi-Modal | OmniLV | 23.23 | 0.6751 | 24.19 | 70.45 | 24.98 | 0.7268 | 11.51 | 69.74 |

Table 17: Restoration on weather/aliasing corruption (rain and ringing). Higher is better for PSNR/SSIM/MUSIQ, lower is better for FID.

| Category | Method | Spatter | | | | SR×2 | | | | SR×4 | | | |
|---|---|---|---|---|---|---|---|---|---|---|---|---|---|
| | | PSNR↑ | SSIM↑ | FID↓ | MUSIQ↑ | PSNR↑ | SSIM↑ | FID↓ | MUSIQ↑ | PSNR↑ | SSIM↑ | FID↓ | MUSIQ↑ |
| Specialized | X-Restormer | – | – | – | – | 19.45 | 0.6572 | 29.84 | 68.14 | 26.19 | 0.8127 | **9.401** | **69.39** |
| | MPRNet | – | – | – | – | – | – | – | – | – | – | – | – |
| | MAXIM | – | – | – | – | – | – | – | – | – | – | – | – |
| All-in-One | X-Restormer | – | – | – | – | 29.29 | 0.9021 | 1.525 | 63.99 | 24.97 | 0.7290 | 21.64 | 38.38 |
| | DA-CLIP | – | – | – | – | – | – | – | – | – | – | – | – |
| | AutoDIR | – | – | – | – | 28.58 | 0.8738 | 4.242 | 57.54 | 24.51 | 0.7214 | 21.37 | 47.27 |
| Visual Prompt | GenLV | 20.45 | 0.5752 | 130.90 | 36.40 | 25.67 | 0.7529 | 18.78 | 40.32 | 25.04 | 0.7070 | 31.85 | 35.15 |
| | PromptGIP | 20.69 | 0.5647 | 116.20 | 35.02 | 23.41 | 0.6585 | 56.96 | 36.74 | 22.53 | 0.6121 | 98.22 | 35.47 |
| | Painter | 18.76 | 0.5392 | 134.00 | 39.12 | 15.37 | 0.5116 | 152.90 | 38.66 | 14.19 | 0.4348 | 187.60 | 33.13 |
| | Prompt-Diffusion | 9.458 | 0.1922 | 185.70 | 59.71 | 9.681 | 0.2657 | 147.40 | 63.40 | 9.546 | 0.2570 | 153.00 | 60.75 |
| Text Prompt | Instruct-Pix2Pix | 15.47 | 0.3732 | 140.00 | 44.72 | 17.15 | 0.5187 | 78.82 | 50.06 | 16.95 | 0.5125 | 87.37 | 43.09 |
| | MGIE | 12.14 | 0.2523 | 214.40 | 57.58 | 12.68 | 0.3487 | 155.30 | 54.03 | 16.80 | 0.5086 | 99.93 | 42.70 |
| | PromptFix | 14.69 | 0.4519 | 215.00 | 59.33 | 28.36 | 0.8741 | 65.52 | 68.68 | 28.03 | 0.8730 | 30.80 | 54.87 |
| | PixWizard | 16.87 | 0.4489 | 93.42 | 68.56 | 19.35 | 0.6113 | 36.45 | 66.59 | 21.00 | 0.5768 | 32.65 | 67.59 |
| Multi-Modal | OmniLV | 23.40 | 0.6696 | 24.00 | 70.13 | 25.33 | 0.7371 | 10.40 | 69.65 | 24.08 | 0.6870 | 12.88 | 69.09 |

Table 18: Restoration on spatter and super-resolution tasks. Higher is better for PSNR/SSIM/MUSIQ, lower is better for FID.

| Category | Method | Brighten (Gamma) | | | | Brighten (Shift) | | | |
|---|---|---|---|---|---|---|---|---|---|
| | | PSNR↑ | SSIM↑ | FID↓ | MUSIQ↑ | PSNR↑ | SSIM↑ | FID↓ | MUSIQ↑ |
| Specialized | Retinexformer | – | – | – | – | – | – | – | – |
| | MPRNet | – | – | – | – | – | – | – | – |
| | MAXIM | – | – | – | – | – | – | – | – |
| All-in-One | X-Restormer | – | – | – | – | – | – | – | – |
| | DA-CLIP | – | – | – | – | – | – | – | – |
| | AutoDIR | – | – | – | – | – | – | – | – |
| Visual Prompt | GenLV | 21.03 | 0.6866 | 41.58 | 40.81 | 21.92 | 0.7040 | 43.45 | 40.11 |
| | PromptGIP | 16.76 | 0.5732 | 81.93 | 37.36 | 15.69 | 0.5436 | 87.07 | 34.04 |
| | Painter | 12.60 | 0.4888 | 155.80 | 37.55 | 12.40 | 0.5071 | 144.80 | 37.21 |
| | Prompt-Diffusion | 9.402 | 0.2540 | 149.90 | 62.72 | 9.410 | 0.2531 | 149.10 | 63.08 |
| Text Prompt | Instruct-Pix2Pix | 13.91 | 0.4975 | 84.05 | 50.29 | 12.76 | 0.4759 | 88.31 | 48.94 |
| | MGIE | 14.86 | 0.5232 | 72.61 | 64.43 | 14.58 | 0.5208 | 65.75 | 62.77 |
| | PromptFix | 11.25 | 0.5163 | 203.30 | 58.17 | 10.27 | 0.4747 | 210.60 | 57.62 |
| | PixWizard | 10.97 | 0.4895 | 87.31 | 64.77 | 11.39 | 0.5015 | 82.71 | 63.50 |
| Multi-Modal | OmniLV | 23.08 | 0.7321 | 15.12 | 70.69 | 22.84 | 0.7145 | 16.57 | 70.36 |

Table 19: Enhancement on brightening tasks (gamma and shift). Higher is better for PSNR/SSIM/MUSIQ, lower is better for FID.

| Category | Method | Contrast Strengthen | | | | Contrast Weaken | | | |
|---|---|---|---|---|---|---|---|---|---|
| | | PSNR↑ | SSIM↑ | FID↓ | MUSIQ↑ | PSNR↑ | SSIM↑ | FID↓ | MUSIQ↑ |
| Specialized | Retinexformer | – | – | – | – | – | – | – | – |
| | MPRNet | – | – | – | – | – | – | – | – |
| | MAXIM | – | – | – | – | – | – | – | – |
| All-in-One | X-Restormer | – | – | – | – | – | – | – | – |
| | DA-CLIP | – | – | – | – | – | – | – | – |
| | AutoDIR | – | – | – | – | – | – | – | – |
| Visual Prompt | GenLV | 21.35 | 0.6561 | 62.12 | 37.91 | 22.80 | 0.7091 | 38.05 | 40.58 |
| | PromptGIP | 16.13 | 0.5214 | 129.70 | 34.65 | 18.49 | 0.5841 | 104.40 | 33.81 |
| | Painter | 10.73 | 0.3605 | 234.90 | 32.25 | 15.38 | 0.5335 | 71.47 | 40.01 |
| | Prompt-Diffusion | 9.503 | 0.2345 | 156.40 | 62.51 | 9.510 | 0.2510 | 156.10 | 63.87 |
| Text Prompt | Instruct-Pix2Pix | 13.30 | 0.4043 | 102.80 | 46.90 | 15.33 | 0.5022 | 86.52 | 54.06 |
| | MGIE | 12.54 | 0.3478 | 114.50 | 57.90 | 15.13 | 0.5081 | 73.18 | 64.48 |
| | PromptFix | 10.45 | 0.3942 | 223.80 | 56.82 | 12.95 | 0.5209 | 190.00 | 57.51 |
| | PixWizard | 13.12 | 0.4472 | 76.24 | 64.76 | 14.76 | 0.4805 | 77.79 | 66.98 |
| Multi-Modal | OmniLV | 21.89 | 0.6635 | 32.22 | 70.24 | 23.58 | 0.7261 | 14.02 | 70.57 |

Table 20: Enhancement on contrast manipulation (strengthen and weaken). Higher is better for PSNR/SSIM/MUSIQ, lower is better for FID.

| Category | Method | Darken (Gamma) | | | | Darken (Shift) | | | |
|---|---|---|---|---|---|---|---|---|---|
| | | PSNR↑ | SSIM↑ | FID↓ | MUSIQ↑ | PSNR↑ | SSIM↑ | FID↓ | MUSIQ↑ |
| Specialized | Retinexformer | 15.81 | 0.6100 | 58.85 | 66.74 | 17.93 | 0.6351 | 52.29 | 66.14 |
| | MPRNet | 16.89 | 0.6916 | 47.41 | 67.45 | 16.28 | 0.6362 | 58.29 | 65.82 |
| | MAXIM | 17.57 | 0.7467 | 39.88 | 69.54 | 14.41 | 0.6210 | 63.38 | 66.54 |
| All-in-One | X-Restormer | – | – | – | – | – | – | – | – |
| | DA-CLIP | 14.97 | 0.5477 | 45.75 | 55.05 | 15.58 | 0.5589 | 41.86 | 54.15 |
| | AutoDIR | 15.62 | 0.6709 | 38.28 | 66.44 | 14.19 | 0.6130 | 38.80 | 67.11 |
| Visual Prompt | GenLV | 21.77 | 0.6865 | 44.23 | 40.15 | 21.92 | 0.6672 | 57.46 | 39.39 |
| | PromptGIP | 18.26 | 0.5605 | 110.90 | 34.57 | 18.28 | 0.5455 | 114.20 | 35.14 |
| | Painter | 13.73 | 0.4745 | 171.80 | 37.91 | 13.82 | 0.4633 | 177.10 | 36.98 |
| | Prompt-Diffusion | 9.323 | 0.2292 | 152.10 | 61.59 | 9.180 | 0.2274 | 159.40 | 60.90 |
| Text Prompt | Instruct-Pix2Pix | 13.15 | 0.3828 | 89.77 | 50.56 | 13.11 | 0.3787 | 88.11 | 50.50 |
| | MGIE | 15.42 | 0.4745 | 68.78 | 63.07 | 15.28 | 0.4642 | 74.68 | 62.25 |
| | PromptFix | 10.39 | 0.3981 | 212.80 | 54.75 | 10.63 | 0.4245 | 206.60 | 56.21 |
| | PixWizard | 13.85 | 0.4452 | 75.34 | 65.44 | 14.30 | 0.4532 | 71.82 | 66.26 |
| Multi-Modal | OmniLV | 21.08 | 0.6821 | 20.92 | 70.25 | 20.38 | 0.6597 | 28.22 | 69.79 |

Table 21: Enhancement on darkening tasks (gamma and shift). Higher is better for PSNR/SSIM/MUSIQ, lower is better for FID.

| Category | Method | Mosaic | | | | Oversharpen | | | |
|---|---|---|---|---|---|---|---|---|---|
| | | PSNR↑ | SSIM↑ | FID↓ | MUSIQ↑ | PSNR↑ | SSIM↑ | FID↓ | MUSIQ↑ |
| Specialized | Retinexformer | – | – | – | – | – | – | – | – |
| | MPRNet | – | – | – | – | – | – | – | – |
| | MAXIM | – | – | – | – | – | – | – | – |
| All-in-One | X-Restormer | – | – | – | – | – | – | – | – |
| | DA-CLIP | – | – | – | – | – | – | – | – |
| | AutoDIR | – | – | – | – | – | – | – | – |
| Visual Prompt | GenLV | 13.46 | 0.5118 | 203.70 | 36.09 | 21.69 | 0.6482 | 113.20 | 37.08 |
| | PromptGIP | 16.93 | 0.5388 | 194.50 | 32.68 | 20.70 | 0.6099 | 101.40 | 38.74 |
| | Painter | 15.08 | 0.5887 | 129.30 | 42.24 | 12.73 | 0.4351 | 201.30 | 38.26 |
| | Prompt-Diffusion | 9.636 | 0.1344 | 246.50 | 55.73 | 9.721 | 0.2542 | 146.70 | 63.76 |
| Text Prompt | Instruct-Pix2Pix | 11.21 | 0.3799 | 118.40 | 51.04 | 15.69 | 0.4556 | 90.60 | 56.01 |
| | MGIE | 9.747 | 0.1919 | 241.70 | 53.35 | 13.53 | 0.3460 | 114.10 | 65.03 |
| | PromptFix | 9.263 | 0.3840 | 192.80 | 52.81 | 14.93 | 0.5861 | 170.70 | 65.83 |
| | PixWizard | 12.43 | 0.3345 | 127.70 | 63.24 | 13.55 | 0.4328 | 75.81 | 71.00 |
| Multi-Modal | OmniLV | 23.93 | 0.7061 | 18.92 | 69.68 | 24.15 | 0.7188 | 17.70 | 71.10 |

Table 22: Enhancement on mosaic artifacts and oversharpening. Higher is better for PSNR/SSIM/MUSIQ, lower is better for FID.

| Category | Method | Saturate Strengthen (HSV) | | | | Saturate Strengthen (YCrCb) | | | |
|---|---|---|---|---|---|---|---|---|---|
| | | PSNR↑ | SSIM↑ | FID↓ | MUSIQ↑ | PSNR↑ | SSIM↑ | FID↓ | MUSIQ↑ |
| Specialized | Retinexformer | – | – | – | – | – | – | – | – |
| | MPRNet | – | – | – | – | – | – | – | – |
| | MAXIM | – | – | – | – | – | – | – | – |
| All-in-One | X-Restormer | – | – | – | – | – | – | – | – |
| | DA-CLIP | – | – | – | – | – | – | – | – |
| | AutoDIR | – | – | – | – | – | – | – | – |
| Visual Prompt | GenLV | 21.52 | 0.6972 | 62.83 | 40.74 | 17.45 | 0.6393 | 67.72 | 39.39 |
| | PromptGIP | 13.60 | 0.4497 | 176.10 | 32.26 | 14.54 | 0.4795 | 182.80 | 33.65 |
| | Painter | 10.80 | 0.3759 | 213.80 | 35.08 | 10.97 | 0.3896 | 221.70 | 34.21 |
| | Prompt-Diffusion | 9.257 | 0.2458 | 154.20 | 63.56 | 9.101 | 0.2431 | 156.30 | 63.65 |
| Text Prompt | Instruct-Pix2Pix | 12.49 | 0.3813 | 115.70 | 50.03 | 12.40 | 0.4078 | 117.40 | 49.70 |
| | MGIE | 12.31 | 0.3957 | 108.70 | 58.81 | 11.67 | 0.3831 | 128.50 | 57.69 |
| | PromptFix | 10.41 | 0.4065 | 221.10 | 56.30 | 10.49 | 0.4137 | 230.80 | 58.30 |
| | PixWizard | 12.78 | 0.4177 | 96.79 | 63.84 | 12.09 | 0.4374 | 102.20 | 64.13 |
| Multi-Modal | OmniLV | 20.91 | 0.6531 | 37.98 | 70.77 | 21.70 | 0.6762 | 35.61 | 71.16 |

Table 23: Enhancement on saturation strengthening in HSV and YCrCb spaces. Higher is better for PSNR/SSIM/MUSIQ, lower is better for FID.

| Category | Method | Saturate Weaken (HSV) | | | | Saturate Weaken (YCrCb) | | | |
|---|---|---|---|---|---|---|---|---|---|
| | | PSNR↑ | SSIM↑ | FID↓ | MUSIQ↑ | PSNR↑ | SSIM↑ | FID↓ | MUSIQ↑ |
| Specialized | Retinexformer | – | – | – | – | – | – | – | – |
| | MPRNet | – | – | – | – | – | – | – | – |
| | MAXIM | – | – | – | – | – | – | – | – |
| All-in-One | X-Restormer | – | – | – | – | – | – | – | – |
| | DA-CLIP | – | – | – | – | – | – | – | – |
| | AutoDIR | – | – | – | – | – | – | – | – |
| Visual Prompt | GenLV | 22.16 | 0.7108 | 53.29 | 40.42 | 22.34 | 0.7123 | 53.33 | 40.57 |
| | PromptGIP | 19.01 | 0.5878 | 140.00 | 36.02 | 19.35 | 0.5982 | 156.10 | 35.99 |
| | Painter | 15.34 | 0.5670 | 134.40 | 38.71 | 15.88 | 0.5673 | 146.10 | 38.60 |
| | Prompt-Diffusion | 9.273 | 0.2497 | 154.60 | 63.33 | 9.390 | 0.2575 | 157.20 | 62.87 |
| Text Prompt | Instruct-Pix2Pix | 16.38 | 0.5168 | 83.61 | 50.96 | 16.76 | 0.5218 | 90.73 | 51.85 |
| | MGIE | 15.23 | 0.5168 | 78.96 | 63.05 | 15.43 | 0.5255 | 90.48 | 62.53 |
| | PromptFix | 13.73 | 0.5779 | 182.50 | 59.53 | 14.59 | 0.5975 | 175.10 | 59.81 |
| | PixWizard | 12.76 | 0.5243 | 91.76 | 65.75 | 13.41 | 0.5287 | 90.64 | 66.78 |
| Multi-Modal | OmniLV | 22.15 | 0.7106 | 25.68 | 70.93 | 22.40 | 0.7149 | 33.78 | 70.74 |

Table 24: Enhancement on saturation weakening in HSV and YCrCb spaces. Higher is better for PSNR/SSIM/MUSIQ, lower is better for FID.

