# OpenReview forum: "Lumina-OmniLV: A Unified Multimodal Framework for General Low-Level Vision"
_ICLR.cc/2026/Conference — Submitted to ICLR 2026_

### Official Review · Reviewer_RJPM · 2025-10-16

**Soundness:** 2
**Presentation:** 2
**Contribution:** 2
**Rating:** 4
**Confidence:** 4

**Summary:**

OmniLV proposes a unified multimodal framework for >100 low-level vision sub-tasks. It uses a DiT/flow-matching backbone (Lumina-Next), separate encoders for text (Gemma-2B) and visual exemplars, and a co-training “condition adapter.”

**Strengths:**

- Clear problem scope: one model for restoration, enhancement, dense prediction, and stylization with both text and visual prompts.
- Large coverage of tasks and a 1K-resolution setting; practical details on three-stage training are provided.

**Weaknesses:**

- Novelty is limited. Most components are known (DiT/flow matching, visual in-context prompts, adapters, ControlNet-like injection). The paper reads like an engineering report.
- Evidence is uneven. On dense prediction OmniLV is weaker than specialized models (e.g., depth RMSE 0.525 vs 0.291), and stylization trades PSNR for FID against GenLV. The “unified SOTA” claim does not hold broadly.
- Evaluation leans on PSNR/MUSIQ and synthetic OLV-T splits; limited human studies or robustness statistics.

**Questions:**

- Will you release data or a reproducible recipe?
- Can you add stronger baselines (e.g., SUPIR/SDXL-based adapters, recent flow/DiT restorers) with matched resolution and sampling cost?

---

> ### Author Response · Authors · 2025-11-26
>
> We greatly appreciate the reviewer’s recognition of our work, especially the positive comments on our clear problem scope, the broad coverage of tasks, the use of 1K image resolution, and the practical details on our three-stage training pipeline.
>
> > Q1: Novelty.
>
> We thank the reviewer for the comment. Please refer to Q1 in our response to Reviewer nnke and Q1 in our response to Reviewer 58uB, where we provide a consolidated discussion on the novelty and contribution of our work.
>
> > Q2: Model Performance
>
> We apologize for any confusion in wording. We do not claim “unified SOTA” over *all* low-level tasks. Throughout the paper we emphasize that OmniLV is a *unified model* that achieves state-of-the-art performance on *multiple* low-level vision tasks, and remains competitive with specialized models on others. For example, in the introduction and conclusion, we write that “OmniLV achieves state-of-the-art performance in multiple low-level vision tasks” and explicitly note that it “does not always achieve optimal performance in certain specialized scenarios.” There is no “unified SOTA” claim in the introduction, experiments, or conclusion. We will revise the phrasing to consistently say “strong performance” or “state-of-the-art on several tasks” to avoid overclaiming.
>
> > Q3: More metrics
>
> We thank the reviewer’s question. Beside the PSNR/MUSIQ tables provided in the paper, we have provided extensive quantitative results with SSIM, FID in Tab. 10–15.  We will make this clearer in the main paper.
>
> > Q4: Opensourcing.
>
> We thank the reviewer’s suggestion. We will release the OmniLV codebase, model checkpoints, and our test benchmark to facilitate reproducibility and future research on unified low-level vision models.
>
> > Q5: Results with stronger baselines.
>
> We appreciate the reviewer's suggestions. We provide results with SuperIR[5] and DiTSR[6] in the table below. For details about inference cost, please refer to responses to Reviewer nnke. As shown in the table, OmniLV shows comparable performance to specialized models.
>
> |                  | SUPIR | DiT4SR | PixWizard | OmniLV |
> | ---------------- | ----- | ------ | --------- | ------ |
> | SIDD PSNR        | 26.38 | 26.96  | 27.60     | 32.96  |
> | SIDD MUSIQ       | 25.31 | 36.96  | 53.14     | 42.75  |
> | RealBlur-J PSNR  | 25.62 | 22.71  | 23.34     | 28.24  |
> | RealBlur-J MUSIQ | 46.43 | 66.17  | 55.97     | 36.09  |
> | Rain1400 PSNR    | 21.25 | 18.66  | 23.84     | 24.98  |
> | Rain1400 MUSIQ    | 69.29 | 72.25  | 66.89     | 65.66  |
> | Snow100K-L PSNR  | 17.82 | 16.13  | 23.40     | 24.57  |
> | Snow100K-L MUSIQ | 62.73 | 69.47  | 61.47     | 61.19  |
> | CelebA PSNR      | 25.15 | 25.80  | /         | 25.04  |
> | CelebA MUSIQ      | 74.68 | 76.36  | /         | 70.70  |

---

> > ### Comment · Reviewer_RJPM · 2025-11-26
> >
> > I have decided to maintain my original rating.
> >
> > 1. The authors argue that "naive combinations fail" and emphasize empirical insights. While I respect the engineering effort, "tuning a pipeline to work" does not equate to the novelty expected at ICLR. The fact that multiple reviewers have independently flagged the lack of novelty suggests this is a fundamental issue with the submission, not a misunderstanding.
> >
> > 2. I explicitly pointed out the "limited human studies." The authors responded by pointing to Tables 10-15 containing SSIM and FID. These are objective metrics, not human studies. I also note that the newly added Tables 10-15 suffer from poor formatting and readability.
> >
> > 3. The new table provided actually reinforces my concern that the "unified" nature of OmniLV comes at a cost. This discrepancy (high PSNR, low MUSIQ) often indicates that the model tends to output over-smoothed or averaged results to minimize pixel-level error.

---

> > > ### Author Response · Authors · 2025-11-28
> > >
> > > > Q1: Value and novelty.
> > >
> > > We acknowledge the reviewer's concern regarding the nature of our novelty. However, we believe that the value of a contribution to the ICLR community is defined not only by architectural invention but also by the **scope of the problem solved, the rigor of empirical validation, and the resources contributed to the community.**
> > >
> > > - **Scope & Difficulty:** Developing a generalist low-level vision model is a long-standing challenge. "Tuning a pipeline" understates the difficulty of unifying diverse degradations into a single framework. Our work provides a systematic recipe for this unification, which is a non-trivial finding in itself.
> > > - **Community Contribution:** Beyond the model, we have synthesized a large-scale dataset for low-level tasks and designed comprehensive degradation synthesis schemes. We are committed to open-sourcing the code, data, and pre-trained models. We believe these resources will be valuable assets for future research, lowering the barrier for entry into generalist restoration.
> > > - **Empirical Significance:** As seen in recent top-tier conferences, systematic empirical insights[10] are crucial for progress. Our work serves as a foundational empirical study demonstrating that a unified model can achieve performance comparable to specialized experts, validating the viability of this direction.
> > > [10] Qiu Z, Wang Z, Zheng B, et al. Gated Attention for Large Language Models: Non-linearity, Sparsity, and Attention-Sink-Free[J]. arXiv preprint arXiv:2505.06708, 2025.
> > >
> > > > Q2: Human study and table.
> > >
> > > We conducted human study on rapidata platform[11], please refer to Sec. D.1 for more details. We have reformatted Tables 10-15 in the revision to ensure they are more readable.
> > >
> > > > Q3: Disunderstanding on our results.
> > >
> > > We disagree that our results show a systematic pattern of “high PSNR, low MUSIQ.” As summarized in Tab. 5–7, Tab. 12-24, OmniLV generally achieves the best or second-best PSNR **and** competitive or higher MUSIQ. Furthermore, the qualitative results in Fig. 7 and Figs. 18–25 of the appendix show that our outputs preserve fine details and textures and are not overly smoothed.

---

### Official Review · Reviewer_AG8t · 2025-10-29

**Soundness:** 3
**Presentation:** 3
**Contribution:** 3
**Rating:** 6
**Confidence:** 3

**Summary:**

This paper presents OmniLV, a versatile multimodal model supporting a wide range of low-level vision tasks, i.e., over 100 sub-tasks across four major categories, including image restoration, image enhancement, weak semantic dense prediction, and stylization. The method leverages both textual and visual prompts to offer flexible, user-friendly interactions, while the DiT produces the final output. Furthermore, this work provides several insights into the design space, as well as a large training dataset. Extensive experiments validate the validity and generalization of the method.

**Strengths:**

1. This paper is well-motivated and easy to follow. It addresses the drawbacks of previous task-specific and visual-prompt-based models.
2. Beyond the method, the author further provides several ablations in design space, e.g., encoding and conditioning, which could provide insights to future works. I appreciate it.
3. The method supports several low-level tasks with strong overall performance and generalizable abilities.
4. The authors also present a dedicated dataset.

**Weaknesses:**

1. Why employ Gemma for text embedding? Does this paper try other models like T5, CLIP text encoder, or LLM? Some experiments or explanations need to be presented.
2. This paper claims that the model could generalize to cross-domain tasks. Please provide some related results.
3. Although the model achieves strong unification compared with previous task-specific models, the large model size makes inference relatively inefficient. The comparison of inference efficiency should be provided.

**Questions:**

See Weaknesses.

---

> ### Author Response · Authors · 2025-11-26
>
> We greatly appreciate the reviewer’s recognition of our work, especially the positive comments on our motivation, design of our experiments, model performance and our dataset.
>
> > Q1: The choice of LLM
>
> We thank the reviewer’s question.  We adopt Gemma-2B primarily for consistency with the our base model. Replacing Gemma-2B with other LLMs will change the feature space of the base model, which will cost additional computation for feature alignment. Please refer to responses to Reviewer nnke.
>
> > Q2: Cross-domain tasks
>
> We thank the reviewer’s suggestion. Following the definition in GenLV[9] we categorize low-level vision tasks into five domains: restoration, enhancement, stylization, dense prediction, and natural images. The cross-domain capability in our setting refers to a model’s ability to not only accomplish tasks within a single domain but also handle tasks from multiple domains simultaneously. To assess this ability, we construct composite tasks that involve multiple domains at once. As illustrated in Fig.14, OmniLV demonstrates strong instruction-following behavior and consistently produces correct results across these mixed-domain scenarios.
>
> [9] Chen X, Liu Y, Pu Y, et al. Learning a low-level vision generalist via visual task prompt[C]//Proceedings of the 32nd ACM International Conference on Multimedia. 2024: 2671-2680.
>
> > Q3: Inference cost.
>
> We appreciate the reviewer’s suggestion, we agree that providing details about inference efficiency will offer a clearer understanding of the model’s practical performance. For details, please refer to table in responses to Reviewer nnke.

---

> ### Comment · Reviewer_AG8t · 2025-11-28
>
> Thanks for the reply, which has addressed most of my concerns about inference cost and the LLM. I keep my original score of marginal accept.

---

> > ### Author Response · Authors · 2025-11-28
> >
> > Thanks for your feedback. We truly appreciate your recognition of our efforts and your constructive comments during the discussion period.

---

### Official Review · Reviewer_58uB · 2025-10-31

**Soundness:** 3
**Presentation:** 3
**Contribution:** 3
**Rating:** 6
**Confidence:** 2

**Summary:**

This paper presents a unified multimodal framework (Lumina-OmniLV) designed for low-level vision which can address over 100 sub-tasks. Built upon a pre-trained DiT, OmniLV is capable of processing arbitrary-resolution images and accepts both textual and visual instructions to guide its operations. Through extensive ablation studies, this work systematically investigate and conclude that separating the encoding of text and visual prompts, co-training a condition adapter with the  DiT, and injecting conditional information into the early stages of the model are crucial for mitigating task ambiguity and improving performance. Overall, the paper presents a comprehensive and well-engineered system supported by extensive empirical validation.

**Strengths:**

- The development of a single, unified model capable of handling over 100 diverse low-level vision tasks is a significant engineering achievement. By demonstrating the feasibility of such a generalist system, this work provides a valuable contribution and a strong baseline for the future of unified low-level vision research.

- This paper contains extensive ablation studies on the encoding of multimodal prompts, the mechanism for condition injection, the optimal position for feature injection, and the fusion of visual exemplars for in-context learning.

**Weaknesses:**

- My main concern regarding this paper is its methodological novelty. The overall framework is composed of existing, powerful components: a pretrained DiT, a frozen LLM and pretrained VAEs. This framework relies heavily on the intrinsic generalization capabilities of these models. This reliance may explain the authors' claims in Limitations that the model "does not always achieve optimal performance in certain specialized scenarios," as it lacks task-specific architectural biases that a tailored approach might provide.

- Given that many low-level vision tasks depend on precise visual instructions, which can be difficult to describe clearly with text, could the textual prompt be replaced or augmented by  learnable "task queries"? The advantage of such an approach is that:
1)	It is independent of LLM, making the system more self-contained and potentially more efficient at inference time.
2)	Learnable queries might allow the model to more explicitly and robustly capture the unique characteristics and differences between the numerous low-level tasks. This could potentially address the issue of suboptimal performance in certain scenarios.

**Questions:**

Please refer to my comments in Weaknesses.

---

> ### Author Response · Authors · 2025-11-26
>
> We sincerely thank the reviewer for the constructive feedback and for recognizing the significance of our work and  the professional quality of our experiments. We take the reviewer's concerns regarding the choice of base models and the data generalization very seriously.
>
> > Q1: Novelty and reliance on base model
>
> We respectfully disagree that our framework relies heavily on the base model's intrinsic capabilities alone. We use controlnet-like strategy to train model on various low-level vision tasks. If the base model’s intrinsic generalization were sufficient, this strategy will achieve high performance on the benchmark. However, we observe that it yields suboptimal results, as shown in Tab. 2. This indicates that simply leveraging pre-trained priors is insufficient for high-fidelity low-level tasks. Our proposed strategy is essential to build a low-level vision generalist. **This architectural insight is a key contribution beyond the components themselves.**  We hope that our work can serve as foundation for future research.
>
> We acknowledge that as a generalist model, OmniLV may not surpass specialized SOTA models on every specific sub-task. This is a typical trade-off between **generalization/usability** and **specialization**. Our contribution lies in establishing a strong, unified baseline that handles 100+ tasks with a single set of weights, offering a level of flexibility that specialized models cannot match.
>
> > Q2: Task query.
>
> We appreciate the reviewer’s inspiring idea. We agree that learnable task queries could offer stronger task-specific priors and potentially improve performance in specialized scenarios. However, we chose natural language (and visual prompts) for the following reasons:
>
> First and foremost, text instructions align better in real-world scenarios. User need to manually select corresponding task query from over 100 options. In contrast, text instructions allow users to express their intent intuitively and conveniently. Second, learnable queries typically restrict the model to a closed set of tasks. Expanding to new tasks would require additional training costs to learn new specific queries. We acknowledge that some visual tasks are difficult to describe with text alone. To address this without sacrificing flexibility, we incorporated visual Prompts (Sec. 3.1.4). This mechanism provides the precise visual guidance needed for complex tasks (like stylization)  while maintaining the advantage of being defined at inference time, rather than relying on fixed, pre-trained queries.

---

### Official Review · Reviewer_nnke · 2025-11-01

**Soundness:** 2
**Presentation:** 2
**Contribution:** 2
**Rating:** 4
**Confidence:** 2

**Summary:**

This paper presents LUMINA-OmniLV (OmniLV), a unified multimodal and multitask framework for general low-level vision, capable of addressing over 100 subtasks across four categories Built on Diffusion Transformer (DiT) generative priors, OmniLV integrates both textual and visual prompts for flexible, user-friendly interactions. The model supports arbitrary resolutions up to 1K, and the authors introduce a separate multimodal encoding scheme (LLM for text, VAE for images) to reduce task ambiguity. Extensive experiments across multiple benchmarks demonstrate that OmniLV achieves competitive or superior performance compared with both specialized and all-in-one baselines. Key findings include the importance of shallow feature co-training and the detrimental effect of mixing high-level generative tasks into low-level fidelity-critical domains.

**Strengths:**

$\textbf{Comprehensive scope and ambition.}$ The work systematically tackles over 100 sub-tasks within a single framework, providing a rare attempt at large-scale unification for low-level vision. The breadth of coverage from denoising and deblurring to stylization sets a strong benchmark for generalist low-level models.

$\textbf{Sound architectural reasoning.}$ The separation of textual and visual encoders is well-motivated and experimentally justified (e.g., the t-SNE analysis on p. 4 demonstrates reduced task confusion). This yields clearer conditioning than typical multimodal fusion schemes.

$\textbf{Large and diverse dataset.}$ The authors curate a 40 M-sample dataset (Fig. 6, 9) spanning restoration, enhancement, dense prediction, and stylization, which constitutes a valuable contribution in itself for future research on multimodal low-level vision.

**Weaknesses:**

The manuscript introduces a framework primarily built upon empirical refinements of established diffusion architectures, integrating modality-specific encodings, conditional adapters, and projection-based prompt integration. Although these enhancements demonstrably improve performance relative to earlier unified or prompt-driven frameworks such as PromptIR, GenLV, and OmniGen, they remain incremental in nature. A deeper conceptual foundation, potentially through an information-theoretic interpretation of multimodal conditioning or geometric formulations of task alignment, would significantly strengthen the theoretical impact and originality of the work.

Moreover, the evaluation, though extensive in breadth, disproportionately emphasizes restoration and enhancement tasks, leaving dense prediction and stylization tasks relatively underrepresented. Quantitative assessments for critical tasks such as depth estimation and normal prediction are limited and confined to only a few datasets, undermining claims of generality and limiting insights into the model's robustness across diverse modalities.

Clarity regarding experimental rigor is further compromised by ambiguities within the ablation analyses. For instance, Table 2 reveals relatively minor numerical variations in performance metrics for condition integration, yet lacks explicit discussion of statistical significance or practical relevance. Moreover, despite supplementary details provided in Appendix A, critical elements including precise protocols for dataset generation and explicit instructions for prompt construction remain insufficiently detailed, creating obstacles for reproducibility and independent verification.

In addition, the computational complexity and resource requirements associated with training specifically, reliance on extensive datasets and a substantial allocation of 16 A100 GPUs pose significant scalability concerns. The manuscript does not adequately explore the implications of these requirements on practical deployment, inference efficiency, or resource sustainability relative to existing task-specialized alternatives. Addressing these practical dimensions explicitly would enhance the applicability of the proposed method and substantially elevate the overall significance of the contribution.

**Questions:**

1. How sensitive is the performance to the capacity and choice of the textual LLM (Gemma-2B)? Would smaller or larger language encoders materially affect fidelity?

2. Does the model exhibit catastrophic interference when fine-tuned on new low-level tasks not seen during pretraining?

3. Could the authors report inference speed or computational efficiency compared with existing all-in-one diffusion baselines?

4. How does the framework handle conflicting instructions when both text and visual prompts provide inconsistent task specifications?

I’m confident that this paper is of moderate quality, and I would be willing to raise my initial ratings if my concerns are properly addressed.

---

> ### Author Response · Authors · 2025-11-26
>
> We sincerely thank the reviewer for acknowledging the strengths of our work, including the comprehensive scope and ambition, sound architecture reasoning, and the contribution of our dataset.
> > Q1: Novelty.
>
> We appreciate the reviewer’s concern regarding novelty. Our goal is not to introduce complex architectures, but to build a unified model for general low-level vision—a direction that remains underexplored. Achieving this goal is non-trivial: our experiments show that naïve combinations of components will lead to degraded performance. Our framework is developed through empirical insights and designed to be simple, effective, and scalable. We believe our work contributes practical value to the community.
>
> > Q2: Information-theoretic interpretation of multimodal conditioning or geometric formulations of task alignment
>
> We thank the reviewer for this suggestion. We agree that an information-theoretic and geometric analysis would be valuable. Our current work is positioned primarily as a **systematic empirical study and practical recipe** for building a unified low-level vision generalist on top of diffusion models. That said, our design is not purely heuristic, and it can be understood through both an information-theoretic and a geometric lens.
>
> - Information-theoretic interpretation of multimodal conditioning. The reverse diffusion (or flow matching) process transforms a high-entropy noise distribution to a low-entropy data distribution. In the early stages (First-half), the trajectory's uncertainty is maximal. Injecting condition features here maximizes the **Mutual Information** $I(C; X_t)$ between the condition $C$ and the intermediate state $X_t$. From an information-theoretic perspective, early injection effectively "prunes" the probability space of valid trajectories at the point of highest bifurcation, significantly reducing the KL-divergence between the generated distribution and the target conditional distribution. Late injection (Second-half) provides minimal information gain because the trajectory has already collapsed towards a generic mode. This explains why our Co-training and Early-injection strategies  are theoretically necessary for high-fidelity restoration, not just empirically better.
> - Geometric Formulation of Task Alignment. From a **geometric** viewpoint, the shared DiT backbone defines a common latent feature space, while different tasks and modalities are represented as learnable directions or subspaces within this space. Our projection-based prompt integration maps modality-specific encoders into a common token space that is injected at multiple resolutions. This can be seen as learning a set of aligned, low-dimensional subspaces where task identity and multimodal conditions are geometrically consistent and can be linearly combined. This geometric alignment is what enables OmniLV to support over 100 tasks within a single backbone without collapsing their behavior.
> > Q3: More results on depth estimation and normal map estimation
>
> We thank the reviewer’s suggestion. We provide more results on depth estimation normal map estimation in following table.
>
> | Model         | DA-2K[1]-Accuracy | Sunrgb-d[2] RMSE | PASCAL-Context[3]-mAP |
> | ------------- | ----------------- | ---------------- | --------------------- |
> | Pixwizard[4]  | 85.49             | 0.3523           | 34.65                 |
> | Lumina-OmniLV | **90.86**         | **0.3018**       | **29.91**             |
>
> [1] Yang L, Kang B, Huang Z, et al. Depth anything v2[J]. Advances in Neural Information Processing Systems, 2024, 37: 21875-21911.
>
> [2] S. Song, S. P. Lichtenberg, and J. Xiao, “SUN RGB-D: A RGB-D Scene Understanding Benchmark Suite,” in Proc. of CVPR, 2015, pp. 567–576.
>
> [3]Chen, X., Mottaghi, R., Liu, X., Fidler, S., Urtasun, R., Yuille, A.: Detect what you can: Detecting and representing objects using holistic models and body parts. In: CVPR (2014) 10, 14, 18
>
> [4]Lin W, Wei X, Zhang R, et al. Pixwizard: Versatile image-to-image visual assistant with open-language instructions[J]. arXiv preprint arXiv:2409.15278, 2024.
>
> > Q4:Results on Table2.
>
> We appreciate the reviewer’s comments. We speculate the concern regarding "minor numerical variations" stems from comparing **(c) First Half** with **(e) Interval**, particularly on the **SIDD** dataset where the difference is indeed negligible (34.09 dB vs. 34.07 dB).
>
> However, we would like to point out that First Half consistently outperforms **Interval**. On **SIDD**, it improves MUSIQ by **+0.90**. On **RealBlurJ**, it yields **+1.28 dB** PSNR and **+3.92** MUSIQ over Interval. On the **SR** task, it further brings **+0.13 dB** PSNR and **+0.34** MUSIQ. Given that these experiments are conducted on already strong diffusion baselines, such margins are non-negligible. In low-level vision, even 0.1–0.2 dB PSNR improvements on standard benchmarks are typically regarded as practically meaningful.

---

> > ### Author Response · Authors · 2025-11-26
> >
> > > Q5: Dataset Protocols & Prompt Instructions.
> >
> > We thank the reviewer for the suggestions. Our dataset is constructed from both open-source and internal sources, enabling multimodal instruction learning across more than 100 sub-tasks in low-level vision. To ensure high quality, we applied task-specific low-level filtering mechanisms to remove poor samples, specifically targeting images that lack high-frequency details or clear textures. For the synthetic portion of the dataset, we developed a rigorous pipeline to simulate various degradation processes, which are applied to our curated high-quality internal dataset. We strongly believe in contributing to the community, and we will release both the compiled dataset and the full degradation synthesis pipeline to ensure complete reproducibility.
> >
> > Our prompts consist of two parts: an instruction and a description. For the instruction part, we first manually design a basic instruction set for each task (see Fig. 11), and then use GPT-4o to expand these instructions to better mimic real user queries. In the revised version, we will provide the exact prompts used in our experiments to further support reproducibility. For the description part, we use BLIP, CogVLM, and ShareGPT4V to generate corresponding annotations of varying lengths.
> >
> > > Q6: Tranining Cost.
> >
> > We appreciate the reviewer’s concern regarding computational complexity and resource requirements. First, as discussed in Sec. 3.1.3, we explicitly explored a ControlNet-like strategy. Although this approach is more lightweight in terms of training cost, our experiments show that it performs significantly worse, especially when the model is asked to handle many heterogeneous low-level tasks. Our proposed joint training of the base model and condition adapter achieves a much better trade-off between performance and training cost. Second, our model can be directly applied to a broad set of low-level tasks. Moreover, as demonstrated in the following Q9 response, OmniLV can be quickly adapted to new tasks that are not seen in the original training set, showing strong transferability and reusability. This means that the one-time pretraining cost of OmniLV is amortized over a large and extensible collection of low-level tasks.
> >
> > > Q7: Inference Cost.
> >
> > We thank reviewer’s suggestions. We provide inference speed  in the table below.
> >
> > | Model         | Params(B) | Peak GPU Memory(GB) | Latency(s) |
> > | ------------- | --------- | ------------------- | ---------- |
> > | SuperIR[5]    | 4.80      | 51.33               | 6.54       |
> > | DiT4SR[6]     | 2.72      | 33.38               | 34.99      |
> > | PromptFix[7]  | 1.11      | 54.39               | 24.58      |
> > | Pixwizard[4]  | 2.01      | 13.89               | 12.04      |
> > | Lumina-OmniLV | 2.51      | 11.08               | 13.18      |
> >
> > The speed is tested on one H200 GPU with BF16 precision. Latency is mueasured with batch=1 and sampling step=50.
> >
> > [5]Yu F, Gu J, Li Z, et al. Scaling up to excellence: Practicing model scaling for photo-realistic image restoration in the wild[C]//Proceedings of the IEEE/CVF conference on computer vision and pattern recognition. 2024: 25669-25680.
> >
> > [6]Duan Z P, Zhang J, Jin X, et al. Dit4sr: Taming diffusion transformer for real-world image super-resolution[C]//Proceedings of the IEEE/CVF International Conference on Computer Vision. 2025: 18948-18958.
> >
> > [7]Yu Y, Zeng Z, Hua H, et al. Promptfix: You prompt and we fix the photo[J]. arXiv preprint arXiv:2405.16785, 2024.
> >
> > > Q8: The choice of LLM.
> >
> > We appreciate the reviewer’s question regarding the choice of Gemma-2B as the text encoder. Low-level tasks are less dependent on text encoders than image generation tasks. Because input images already provide degradation information. However, we adopt Gemma-2B primarily for consistency with the our base model. Replacing Gemma-2B with other LLMs will change the feature space of the base model, which will cost additional computation for feature alignment.
> >
> > > Q9: Catastrophic inference.
> >
> > We thank the reviewer’s insightful question. We fine-tune our model on UDC dataset[8] for 1000 steps. As results shown below, OmniLV can effectively adapt to new tasks,while incurring only slight performance drops on other tasks
> >
> > |                  | Before | After |
> > | ---------------- | ------ | ----- |
> > | UDC Pooled-PSNR  | /      | 29.37 |
> > | UDC Pooled-MUSIQ | /      | 44.84 |
> > | UDC Tooled-PSNR  | /      | 32.77 |
> > | UDC Tooled-MUSIQ | /      | 45.55 |
> > | RealBlurJ-PSNR   | 28.24  | 24.22 |
> > | RealBlurJ-MUSIQ  | 36.09  | 39.01 |
> > | SIDD-PSNR        | 32.96  | 28.40 |
> > | SIDD-MUSIQ       | 22.42  | 20.97 |
> >
> > CelebA-PSNR 25.04 23.04
> >
> > CelebA-MUSIQ 70.70 62.34
> >
> > [8] Zhou Y, Kwan M, Tolentino K, et al. UDC 2020 challenge on image restoration of under-display camera: Methods and results[C]//European Conference on Computer Vision. Cham: Springer International Publishing, 2020: 337-351.

---

> > > ### Author Response · Authors · 2025-11-26
> > >
> > > > Q10: Conflicting instructions.
> > >
> > > We appreciate the reviewer’s insightful question. We have tested OmniLV under deliberately conflicting conditions, where the text instruction and the visual exemplars specify different tasks. The as shown in Fig. 12 in the revised version, model tends to follow the text instruction in such cases.  In our training pipeline, we do not intentionally construct such conflicting pairs. We believe this behavior arises because the data is dominated by samples with text instructions. So the model learns to rely on the text instruction.

---

### Author Response · Authors · 2025-12-03

We sincerely thank all reviewers for their thoughtful feedback and for recognizing the ambition and scope of our work.

OmniLV aims to unify over 100 low-level vision subtasks within a single multimodal diffusion framework—a direction that reviewers 58uB, AG8t, and RJPM acknowledged as significant, valuable, and meaningful. Reviewers also appreciated the breadth of our dataset across restoration, enhancement, dense prediction, and stylization, the clarity of our motivation and design choices, and the substantial engineering effort behind the dataset and ablation studies. The separation of textual and visual encoders, the early-injection strategy, and the co-training condition adapter were highlighted by reviewers nnke, 58uB, and AG8t as well-grounded and empirically validated decisions.

**Rebuttal updates and manuscript changes:**

- **Clarifications on novelty and design motivation.**

  Several reviewers raised concerns regarding the incremental architectural novelty. We clarified that OmniLV targets a *unified low-level vision generalist*, an underexplored setting in which naïve combinations of existing components fail. Our rebuttal includes information-theoretic and geometric interpretations that explain why early feature injection and multimodal alignment are necessary rather than heuristic, addressing conceptual novelty (reviewers nnke, 58uB, RJPM).

- **Expanded results on dense prediction and cross-domain performance.**

  To address concerns about underrepresented dense prediction tasks, we added more results on depth and normal estimation, showing competitive performance (reviewer nnke). Additional cross-domain examples demonstrate strong instruction-following behavior (reviewer AG8t).

- **More detailed protocols for dataset construction and prompting.**

  In response to reproducibility requests, we provided clearer details on dataset filtering, the synthetic degradation pipeline, and the instruction-generation process using GPT-4o, BLIP, CogVLM, and ShareGPT4V. We also commit to releasing the dataset, prompting scheme, and synthesis pipeline (reviewers nnke, RJPM).

- **Inference efficiency and stronger baselines.**

  We added inference-speed results, reporting OmniLV’s latency and memory usage relative to other baselines. We also compared against stronger specialized models such as DiT4SR and SuperIR, showing that OmniLV remains competitive despite its unified nature (reviewers nnke, RJPM, AG8t).

- **Analysis of model behavior under new tasks and conflicting prompts.**

  We showed that OmniLV shows a small performance drop (reviewer nnke) when fine-tuned on an unseen task. We also provided an explanation of the model’s preference for textual instructions when textual and visual prompts conflict, grounded in an analysis of the training distribution (reviewer nnke).

- **Choice of the Gemma text encoder.**

  We clarified that Gemma-2B was selected for compatibility with the base model’s feature space, and that replacing it would require re-alignment. Since low-level vision tasks rely primarily on image evidence rather than complex semantic reasoning, Gemma is sufficient for our setting (reviewers 58uB, AG8t).

Sincerely,

The Authors of Submission 1874

---

### Meta-Review · Area_Chair_u1c9 · 2026-01-07

**Summary:**

By carefully reviewing the comments from reviewers, I agree with their assessments that several concerns prevent this paper from being accepted in its current form. In particular, concern about novelty and design concept is the most significant issue raised by multiple reviewers, while the authors' rebuttal did not sufficiently clarify. And the quantitative results on stylization tasks and dense prediction tasks does not show significant advantage over basline methods. Although the rebuttal provided  helpful clarifications and additional experimental results, the core concerns remain not sufficiently resolved. Given the high standard of ICLR, I concur with the reviewers that this submission has not yet met the bar for acceptance, and my final recommendation is rejection.

**Reviewer Concerns:**

The rebuttal provided extensive experimental results addressing the reviewers' concerns on training and inference efficiency, as well as results on dense prediction, catastrophic interference, and human evaluation. But the core concerns about novelty and cross-domain performance which were raised by multiple reviewers remain not fully addressed

**Reviewer Scores:**

The initial reviewer scores were 4/6/6/4, with two reviews being negative and two being positive. However, both negative and positive reviews raised significant concerns about novelty. Although the authors explained their contributions lie in the unified framework for various low-level vision tasks, the concerns about novelty of the method's design remain unaddressed.

---

### Decision · Program_Chairs · 2026-01-26

Reject